# Deep learning for symbolic mathematics

**Guillaume Lample**[*]
Facebook AI Research
glample@fb.com

**François Charton**[*]
Facebook AI Research
fcharton@fb.com

## Abstract

Neural networks have a reputation for being better at solving statistical or approximate problems than at performing calculations or working with symbolic data. In this paper, we show that they can be surprisingly good at more elaborated tasks in mathematics, such as symbolic integration and solving differential equations. We propose a syntax for representing mathematical problems, and methods for generating large datasets that can be used to train sequence-to-sequence models. We achieve results that outperform commercial Computer Algebra Systems such as Matlab or Mathematica.

## 1 Introduction

A longstanding tradition in machine learning opposes rule-based inference to statistical learning (Rumelhart et al., 1986), and neural networks clearly stand on the statistical side. They have proven to be extremely effective in statistical pattern recognition and now achieve state-of-the-art performance on a wide range of problems in computer vision, speech recognition, natural language processing (NLP), etc. However, the success of neural networks in symbolic computation is still extremely limited: combining symbolic reasoning with continuous representations is now one of the challenges of machine learning.

Only a few studies investigated the capacity of neural network to deal with mathematical objects, and apart from a small number of exceptions (Zaremba et al., 2014; Loos et al., 2017; Allamanis et al., 2017; Arabshahi et al., 2018b), the majority of these works focus on arithmetic tasks like integer addition and multiplication (Zaremba & Sutskever, 2014; Kaiser & Sutskever, 2015; Trask et al., 2018). On these tasks, neural approaches tend to perform poorly, and require the introduction of components biased towards the task at hand (Kaiser & Sutskever, 2015; Trask et al., 2018).

In this paper, we consider mathematics, and particularly symbolic calculations, as a target for NLP models. More precisely, we use sequence-to-sequence models (seq2seq) on two problems of symbolic mathematics: function integration and ordinary differential equations (ODEs). Both are difficult, for trained humans and computer software. For integration, humans are taught a set of rules (integration by parts, change of variable, etc.), that are not guaranteed to succeed, and Computer Algebra Systems use complex algorithms (Geddes et al., 1992) that explore a large number of specific cases. For instance, the complete description of the Risch algorithm (Risch, 1970) for function integration is more than 100 pages long.

Yet, function integration is actually an example where pattern recognition should be useful: detecting that an expression is of the form $yy'(y^2 + 1)^{-1/2}$ suggests that its primitive will contain $\sqrt{y^2 + 1}$. Detecting this pattern may be easy for small expressions $y$, but becomes more difficult as the number of operators in $y$ increases. However, to the best of our knowledge, no study has investigated the ability of neural networks to detect patterns in mathematical expressions.

We first propose a representation of mathematical expressions and problems that can be used by seq2seq models, and discuss the size and structure of the resulting problem space. Then, we show how to generate datasets for supervised learning of integration and first and second order differential equations. Finally, we apply seq2seq models to these datasets, and show that they achieve a better performance than state-of-the-art computer algebra programs, namely Matlab and Mathematica.

---

[*] Equal contribution.

## 2 MATHEMATICS AS A NATURAL LANGUAGE

### 2.1 EXPRESSIONS AS TREES

Mathematical expressions can be represented as trees, with operators and functions as internal nodes, operands as children, and numbers, constants and variables as leaves. The following trees represent expressions $2 + 3 \times (5 + 2)$, $3x^2 + \cos(2x) - 1$, and $\frac{\partial^2 \psi}{\partial x^2} - \frac{1}{\nu^2} \frac{\partial^2 \psi}{\partial t^2}$:

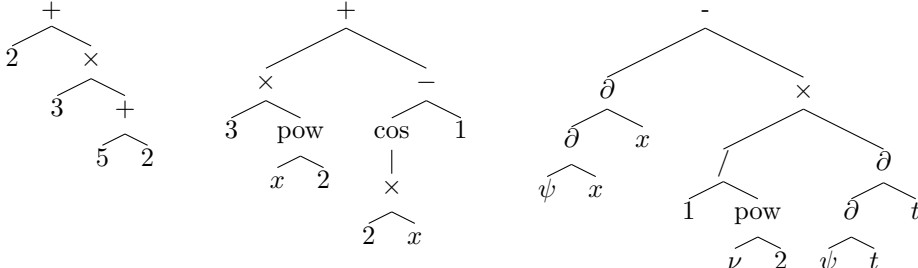

Trees disambiguate the order of operations, take care of precedence and associativity and eliminate the need for parentheses. Up to the addition of meaningless symbols like spaces, punctuation or redundant parentheses, different expressions result in different trees. With a few assumptions, discussed in Section A of the appendix, there is a one-to-one mapping between expressions and trees.

We consider expressions as sequences of mathematical symbols. $2 + 3$ and $3 + 2$ are different expressions, as are $\sqrt{4}x$ and $2x$, and they will be represented by different trees. Most expressions represent meaningful mathematical objects. $x / 0$, $\sqrt{-2}$ or $\log(0)$ are also legitimate expressions, even though they do not necessarily make mathematical sense.

Since there is a one-to-one correspondence between trees and expressions, equality between expressions will be reflected over their associated trees, as an equivalence : since $2+3 = 5 = 12-7 = 1 \times 5$, the four trees corresponding to these expressions are equivalent.

Many problems of formal mathematics can be reframed as operations over expressions, or trees. For instance, expression simplification amounts to finding a shorter equivalent representation of a tree. In this paper, we consider two problems: symbolic integration and differential equations. Both boil down to transforming an expression into another, e.g. mapping the tree of an equation to the tree of its solution. We regard this as a particular instance of machine translation.

### 2.2 TREES AS SEQUENCES

Machine translation systems typically operate on sequences (Sutskever et al., 2014; Bahdanau et al., 2015). Alternative approaches have been proposed to generate trees, such as Tree-LSTM (Tai et al., 2015) or Recurrent Neural Network Grammars (RNNG) (Dyer et al., 2016; Eriguchi et al., 2017). However, tree-to-tree models are more involved and much slower than their seq2seq counterparts, both at training and at inference. For the sake of simplicity, we use seq2seq models, which were shown to be effective at generating trees, e.g. in the context of constituency parsing (Vinyals et al., 2015), where the task is to predict a syntactic parse tree of input sentences.

Using seq2seq models to generate trees requires to map trees to sequences. To this effect, we use prefix notation (also known as normal Polish notation), writing each node before its children, listed from left to right. For instance, the arithmetic expression $2 + 3 * (5 + 2)$ is represented as the sequence $[+ \ 2 \ * \ 3 \ + \ 5 \ 2]$. In contrast to the more common infix notation $2 + 3 * (5 + 2)$, prefix sequences need no parentheses and are therefore shorter. Inside sequences, operators, functions or variables are represented by specific tokens, and integers by sequences of digits preceded by a sign. As in the case between expressions and trees, there exists a one-to-one mapping between trees and prefix sequences.

### 2.3 GENERATING RANDOM EXPRESSIONS

To create training data, we need to generate sets of random mathematical expressions. However, sampling uniformly expressions with $n$ internal nodes is not a simple task. Naive algorithms (such as

recursive methods or techniques using fixed probabilities for nodes to be leaves, unary, or binary) tend to favour deep trees over broad trees, or left-leaning over right leaning trees. Here are examples of different trees that we want to generate with the same probability.

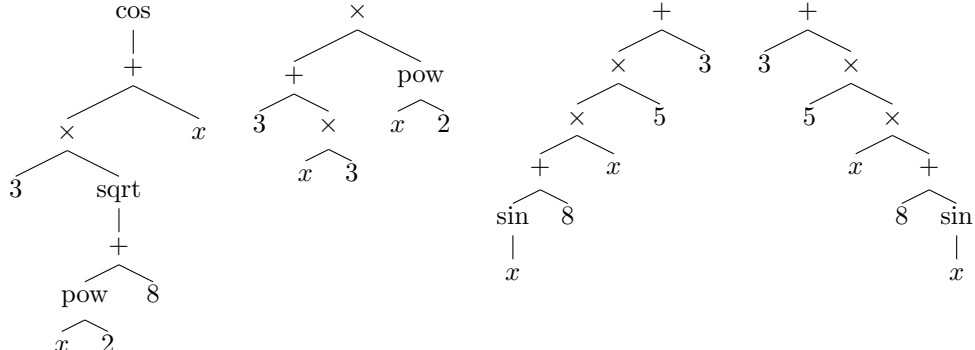

In Section C of the appendix, we present an algorithm to generate random trees and expressions, where the four expression trees above are all generated with the same probability.

## 2.4 COUNTING EXPRESSIONS

We now investigate the number of possible expressions. Expressions are created from a finite set of variables (i.e. literals), constants, integers, and a list of operators that can be simple functions (e.g. cos or exp) or more involved operators (e.g. differentiation or integration). More precisely, we define our problem space as:

- trees with up to $n$ internal nodes
- a set of $p_1$ unary operators (e.g. $\cos, \sin, \exp, \log$)
- a set of $p_2$ binary operators (e.g. $+, -, \times, \text{pow}$)
- a set of $L$ leaf values containing variables (e.g. $x, y, z$), constants (e.g. $e, \pi$), integers (e.g. $\{-10, \dots, 10\}$)

If $p_1 = 0$, expressions are represented by binary trees. The number of binary trees with $n$ internal nodes is given by the $n$-th Catalan numbers $C_n$ (Sloane, 1996). A binary tree with $n$ internal nodes has exactly $n + 1$ leaves. Each node and leaf can take respectively $p_2$ and $L$ different values. As a result, the number of expressions with $n$ binary operators can be expressed by:

$$E_n = C_n p_2^n L^{n+1} \approx \frac{4^n}{n\sqrt{\pi n}} p_2^n L^{n+1} \quad \text{with} \quad C_n = \frac{1}{n+1}\binom{2n}{n}$$

If $p_1 > 0$, expressions are unary-binary trees, and the number of trees with $n$ internal nodes is the $n$-th large Schroeder number $S_n$ (Sloane, 1996). It can be computed by recurrence using the following equation:

$$(n+1)S_n = 3(2n-1)S_{n-1} - (n-2)S_{n-2} \tag{1}$$

Finally, the number $E_n$ of expressions with $n$ internal nodes, $p_1$ unary operator, $p_2$ binary operators and $L$ possible leaves is recursively computed as

$$(n+1)E_n = (p_1 + 2Lp_2)(2n-1)E_{n-1} - p_1(n-2)E_{n-2} \tag{2}$$

If $p_1 = p_2 = L = 1$, Equation 2 boils down to Equation 1. If $p_2 = L = 1, p_1 = 0$, we have $(n+1)E_n = 2(2n-1)E_{n-1}$ which is the recurrence relation satisfied by Catalan numbers. The derivations and properties of all these formulas are provided in Section B of the appendix.

In Figure 1, we represent the number of binary trees ($C_n$) and unary-binary trees ($S_n$) for different numbers of internal nodes. We also represent the number of possible expressions ($E_n$) for different sets of operators and leaves.

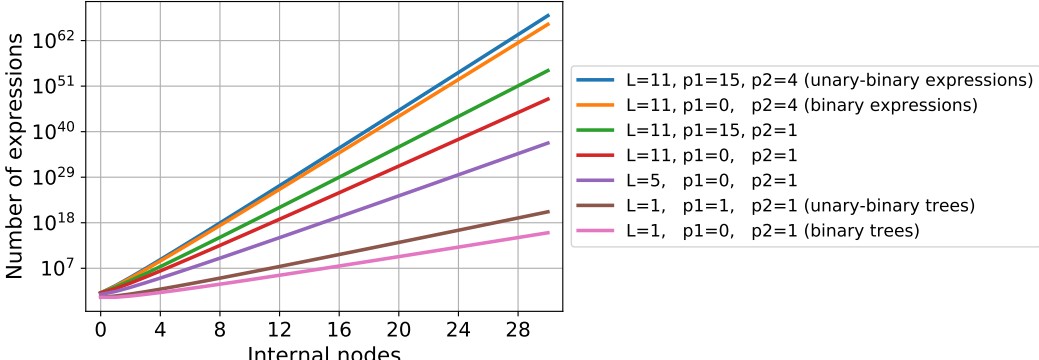

Figure 1: **Number of trees and expressions for different numbers of operators and leaves.** $p_1$ and $p_2$ correspond to the number of unary and binary operators respectively, and $L$ to the number of possible leaves. The bottom two curves correspond to the number of binary and unary-binary trees (enumerated by Catalan and Schroeder numbers respectively). The top two curves represent the associated number of expressions. We observe that adding leaves and binary operators significantly increases the size of the problem space.

## 3 GENERATING DATASETS

Having defined a syntax for mathematical problems and techniques to randomly generate expressions, we are now in a position to build the datasets our models will use. In the rest of the paper, we focus on two problems of symbolic mathematics: function integration and solving ordinary differential equations (ODE) of the first and second order.

To train our networks, we need datasets of problems and solutions. Ideally, we want to generate representative samples of the problem space, i.e. randomly generate functions to be integrated and differential equations to be solved. Unfortunately, solutions of random problems sometimes do not exist (e.g. the integrals of $f(x) = \exp(x^2)$ or $f(x) = \log(\log(x))$ cannot be expressed with usual functions), or cannot be easily derived. In this section, we propose techniques to generate large training sets for integration and first and second order differential equations.

### 3.1 INTEGRATION

We propose three approaches to generate functions with their associated integrals.

**Forward generation (FWD).** A straightforward approach is to generate random functions with up to $n$ operators (using methods from Section 2) and calculate their integrals with a computer algebra system. Functions that the system cannot integrate are discarded. This generates a representative sample of the subset of the problem space that can be successfully solved by an external symbolic mathematical framework.

**Backward generation (BWD).** An issue with the forward approach is that the dataset only contains functions that symbolic frameworks can solve (they sometimes fail to compute the integral of integrable functions). Also, integrating large expressions is time expensive, which makes the overall method particularly slow. Instead, the backward approach generates a random function $f$, computes its derivative $f'$, and adds the pair $(f', f)$ to the training set. Unlike integration, differentiation is always possible and extremely fast even for very large expressions. As opposed to the forward approach, this method does not depend on an external symbolic integration system.

**Backward generation with integration by parts (IBP).** An issue with the backward approach is that it is very unlikely to generate the integral of simple functions like $f(x) = x^3 \sin(x)$. Its integral, $F(x) = -x^3 \cos(x) + 3x^2 \sin(x) + 6x \cos(x) - 6 \sin(x)$, a function with 15 operators, has a very low probability of being generated randomly. Besides, the backward approach tends to generate examples where the integral (the solution) is shorter than the derivative (the problem), while forward generation favors the opposite (see Figure 2 in Section E in the Appendix). To address this issue, we

leverage integration by parts: given two randomly generated functions $F$ and $G$, we compute their respective derivatives $f$ and $g$. If $fG$ already belongs to the training set, we know its integral, and we can compute the integral of $Fg$ as:

$$\int Fg = FG - \int fG$$

Similarly, if $Fg$ is in the training set, we can infer the integral of $fG$. Whenever we discover the integral of a new function, we add it to the training set. If none of $fG$ or $Fg$ are in the training set, we simply generate new functions $F$ and $G$. With this approach, we can generate the integrals of functions like $x^{10}\sin(x)$ without resorting to an external symbolic integration system.

**Comparing different generation methods.** Table 1 in Section 4.1 summarizes the differences between the three generation methods. The `FWD` method tends to generate short problems with long solutions (that computer algebras can solve). The `BWD` approach, on the other hand, generates long problems with short solutions. `IBP` generates datasets comparable to `FWD` (short problems and long solutions), without an external computer algebra system. A mixture of `BWD` and `IBP` generated data should therefore provide a better representation of problem space, without resorting to external tools. Examples of functions / integrals for the three approaches are given in Table 9 of the Appendix.

## 3.2 FIRST ORDER DIFFERENTIAL EQUATION (ODE 1)

We now present a method to generate first order differential equations with their solutions. We start from a bivariate function $F(x, y)$ such that the equation $F(x, y) = c$ (where $c$ is a constant) can be analytically solved in $y$. In other words, there exists a bivariate function $f$ that satisfies $\forall(x, c), F\big(x, f(x, c)\big) = c$. By differentiation with respect to $x$, we have that $\forall x, c$:

$$\frac{\partial F\big(x, f_c(x)\big)}{\partial x} + f_c'(x)\frac{\partial F\big(x, f_c(x)\big)}{\partial y} = 0$$

where $f_c = x \mapsto f(x, c)$. As a result, for any constant $c$, $f_c$ is solution of the first order differential equation:

$$\frac{\partial F\big(x, y\big)}{\partial x} + y'\frac{\partial F\big(x, y\big)}{\partial y} = 0 \tag{3}$$

With this approach, we can use the method described in Section C of the appendix to generate arbitrary functions $F(x, y)$ analytically solvable in $y$, and create a dataset of differential equations with their solutions.

Instead of generating a random function $F$, we can generate a solution $f(x, c)$, and determine a differential equation that it satisfies. If $f(x, c)$ is solvable in $c$, we compute $F$ such that $F\big(x, f(x, c)\big) = c$. Using the above approach, we show that for any constant $c$, $x \mapsto f(x, c)$ is a solution of differential Equation 3. Finally, the resulting differential equation is factorized, and we remove all positive factors from the equation.

A necessary condition for this approach to work is that the generated functions $f(x, c)$ can be solved in $c$. For instance, the function $f(x, c) = c \times \log(x + c)$ cannot be analytically solved in $c$, i.e. the function $F$ that satisfies $F\big(x, f(x, c)\big) = c$ cannot be written with usual functions. Since all the operators and functions we use are invertible, a simple condition to ensure the solvability in $c$ is to guarantee that $c$ only appears once in the leaves of the tree representation of $f(x, c)$. A straightforward way to generate a suitable $f(x, c)$ is to sample a random function $f(x)$ by the methods described in Section C of the appendix, and to replace one of the leaves in its tree representation by $c$. Below is an example of the whole process:

| | |
|---|---|
| Generate a random function | $f(x) = x\log(c\,/\,x)$ |
| Solve in $c$ | $c = xe^{\frac{f(x)}{x}} = F(x, f(x))$ |
| Differentiate in $x$ | $e^{\frac{f(x)}{x}}\big(1 + f'(x) - \frac{f(x)}{x}\big) = 0$ |
| Simplify | $xy' - y + x = 0$ |

### 3.3 SECOND ORDER DIFFERENTIAL EQUATION (ODE 2)

Our method for generating first order equations can be extended to the second order, by considering functions of three variables $f(x, c_1, c_2)$ that can be solved in $c_2$. As before, we derive a function of three variables $F$ such that $F\big(x, f(x, c_1, c_2), c_1\big) = c_2$. Differentiation with respect to $x$ yields a first order differential equation:

$$\frac{\partial F(x, y, c_1)}{\partial x} + f'_{c_1, c_2}(x) \frac{\partial F(x, y, c_1)}{\partial y}\bigg|_{y = f_{c_1, c_2}(x)} = 0$$

where $f_{c_1, c_2} = x \mapsto f(x, c_1, c_2)$. If this equation can be solved in $c_1$, we can infer another three-variable function $G$ satisfying $\forall x, G\big(x, f_{c_1, c_2}(x), f'_{c_1, c_2}(x)\big) = c_1$. Differentiating with respect to $x$ a second time yields the following equation:

$$\frac{\partial G(x, y, z)}{\partial x} + f'_{c_1, c_2}(x) \frac{\partial G(x, y, z)}{\partial y} + f''_{c_1, c_2}(x) \frac{\partial G(x, y, z)}{\partial z}\bigg|_{\substack{y = f_{c_1, c_2}(x) \\ z = f'_{c_1, c_2}(x)}} = 0$$

Therefore, for any constants $c_1$ and $c_2$, $f_{c_1, c_2}$ is solution of the second order differential equation:

$$\frac{\partial G(x, y, y')}{\partial x} + y' \frac{\partial G(x, y, y')}{\partial y} + y'' \frac{\partial G(x, y, y')}{\partial z} = 0$$

Using this approach, we can create pairs of second order differential equations and solutions, provided we can generate $f(x, c_1, c_2)$ is solvable in $c_2$, and that the corresponding first order differential equation is solvable in $c_1$. To ensure the solvability in $c_2$, we can use the same approach as for first order differential equation, e.g. we create $f_{c_1, c_2}$ so that $c_2$ has exactly one leaf in its tree representation. For $c_1$, we employ a simple approach where we simply skip the current equation if we cannot solve it in $c_1$. Although naive, we found that the differentiation equation can be solved in $c_1$ about $50\%$ the time. As an example:

| | |
|---|---|
| Generate a random function | $f(x) = c_1 e^x + c_2 e^{-x}$ |
| Solve in $c_2$ | $c_2 = f(x)e^x - c_1 e^{2x} = F(x, f(x), c_1)$ |
| Differentiate in $x$ | $e^x\big(f'(x) + f(x)\big) - 2c_1 e^{2x} = 0$ |
| Solve in $c_1$ | $c_1 = \dfrac{1}{2}e^{-x}\big(f'(x) + f(x)\big) = G(x, f(x), f'(x))$ |
| Differentiate in $x$ | $0 = \dfrac{1}{2}e^{-x}\big(f''(x) - f(x)\big)$ |
| Simplify | $y'' - y = 0$ |

### 3.4 DATASET CLEANING

**Equation simplification**   In practice, we simplify generated expressions to reduce the number of unique possible equations in the training set, and to reduce the length of sequences. Also, we do not want to train our model to predict $x + 1 + 1 + 1 + 1 + 1$ when it can simply predict $x + 5$. As a result, sequences $[+\ 2\ +\ x\ 3]$ and $[+\ 3\ +\ 2\ x]$ will both be simplified to $[+\ x\ 5]$ as they both represent the expression $x + 5$. Similarly, the expression $\log(e^{x+3})$ will be simplified to $x + 3$, the expression $\cos^2(x) + \sin^2(x)$ will be simplified to 1, etc. On the other hand, $\sqrt{(x-1)^2}$ will not be simplified to $x - 1$ as we do not make any assumption on the sign of $x - 1$.

**Coefficients simplification**   In the case of first order differential equations, we modify generated expressions by equivalent expressions up to a change of variable. For instance, $x + x\tan(3) + cx + 1$ will be simplified to $cx + 1$, as a particular choice of the constant $c$ makes these two expressions identical. Similarly, $\log(x^2) + c\log(x)$ becomes $c\log(x)$.

We apply a similar technique for second order differential equations, although simplification is sometimes a bit more involved because there are two constants $c_1$ and $c_2$. For instance, $c_1 - c_2 x/5 + c_2 + 1$ is simplified to $c_1 x + c_2$, while $c_2 e^{c_1} e^{c_1 x e - 1}$ can be expressed with $c_2 e^{c_1 x}$, etc.

We also perform transformations that are not strictly equivalent, as long as they hold under specific assumptions. For instance, we simplify $\tan(\sqrt{c_2} x) + \cosh(c_1 + 1) + 4$ to $c_1 + \tan(c_2 x)$, although the constant term can be negative in the second expression, but not the first one. Similarly $e^3 e^{c_1 x} e^{c_1 \log(c_2)}$ is transformed to $c_2 e^{c_1 x}$.

**Invalid expressions**  Finally, we also remove invalid expressions from our dataset. For instance, expressions like $\log(0)$ or $\sqrt{-2}$. To detect them, we compute in the expression tree the values of subtrees that do not depend on $x$. If a subtree does not evaluate to a finite real number (e.g. $-\infty$, $+\infty$ or a complex number), we discard the expression.

## 4 EXPERIMENTS

### 4.1 DATASET

For all considered tasks, we generate datasets using the method presented in Section 3, with:

- expressions with up to $n = 15$ internal nodes
- $L = 11$ leaf values in $\{x\} \cup \{-5, \ldots, 5\} \setminus \{0\}$
- $p_2 = 4$ binary operators: $+, -, \times, /$
- $p_1 = 15$ unary operators: exp, log, sqrt, sin, cos, tan, $\sin^{-1}$, $\cos^{-1}$, $\tan^{-1}$, sinh, cosh, tanh, $\sinh^{-1}$, $\cosh^{-1}$, $\tanh^{-1}$

Statistics about our datasets are presented in Table 1. As discussed in Section 3.1, we observe that the backward approach generates derivatives (i.e. inputs) significantly longer than the forward generator. We discuss this in more detail in Section E of the appendix.

|  | Forward | Backward | Integration by parts | ODE 1 | ODE 2 |
|---|---|---|---|---|---|
| Training set size | 20M | 40M | 20M | 40M | 40M |
| Input length | $18.9_{\pm 6.9}$ | $70.2_{\pm 47.8}$ | $17.5_{\pm 9.1}$ | $123.6_{\pm 115.7}$ | $149.1_{\pm 130.2}$ |
| Output length | $49.6_{\pm 48.3}$ | $21.3_{\pm 8.3}$ | $26.4_{\pm 11.3}$ | $23.0_{\pm 15.2}$ | $24.3_{\pm 14.9}$ |
| Length ratio | 2.7 | 0.4 | 2.0 | 0.4 | 0.1 |
| Input max length | 69 | 450 | 226 | 508 | 508 |
| Output max length | 508 | 75 | 206 | 474 | 335 |

Table 1: **Training set sizes and length of expressions (in tokens) for different datasets.** `FWD` and `IBP` tend to generate examples with outputs much longer than the inputs, while the `BWD` approach generates shorter outputs. Like in the `BWD` case, ODE generators tend to produce solutions much shorter than their equations.

### 4.2 MODEL

For all our experiments, we train a seq2seq model to predict the solutions of given problems, i.e. to predict a primitive given a function, or predict a solution given a differential equation. We use a transformer model (Vaswani et al., 2017) with 8 attention heads, 6 layers, and a dimensionality of 512. In our experiences, using larger models did not improve the performance. We train our models with the Adam optimizer (Kingma & Ba, 2014), with a learning rate of $10^{-4}$. We remove expressions with more than 512 tokens, and train our model with 256 equations per batch.

At inference, expressions are generated by a beam search (Koehn, 2004; Sutskever et al., 2014), with early stopping. We normalize the log-likelihood scores of hypotheses in the beam by their sequence length. We report results with beam widths of 1 (i.e. greedy decoding), 10 and 50.

During decoding, nothing prevents the model from generating an invalid prefix expression, e.g. [+ 2 ∗ 3 ]. To address this issue, Dyer et al. (2016) use constraints during decoding, to ensure

that generated sequences can always be converted to valid expression trees. In our case, we found that model generations are almost always valid and we do not use any constraint. When an invalid expression is generated, we simply consider it as an incorrect solution and ignore it.

## 4.3 EVALUATION

At the end of each epoch, we evaluate the ability of the model to predict the solutions of given equations. In machine translation, hypotheses given by the model are compared to references written by human translators, typically with metrics like the BLEU score (Papineni et al., 2002) that measure the overlap between hypotheses and references. Evaluating the quality of translations is a very difficult problem, and many studies showed that a better BLEU score does not necessarily correlate with a better performance according to human evaluation. Here, however, we can easily verify the correctness of our model by simply comparing generated expressions to their reference solutions.

For instance, for the given differential equation $xy' - y + x = 0$ with a reference solution $x \log(c\,/\,x)$ (where $c$ is a constant), our model may generate $x \log(c) - x \log(x)$. We can check that these two solutions are equal, although they are written differently, using a symbolic framework like SymPy (Meurer et al., 2017).

However, our model may also generate $xc - x \log(x)$ which is also a valid solution, that is actually equivalent to the previous one for a different choice of constant $c$. In that case, we replace $y$ in the differential equation by the model hypothesis. If $xy' - y + x = 0$, we conclude that the hypothesis is a valid solution. In the case of integral computation, we can simply differentiate the model hypothesis, and compare it with the function to integrate. For the three problems, we measure the accuracy of our model on equations from the test set.

Since we can easily verify the correctness of generated expressions, we consider all hypotheses in the beam, and not only the one with the highest score. We verify the correctness of each hypothesis, and consider that the model successfully solved the input equation if one of them is correct. As a result, results with "Beam size 10" indicate that at least one of the 10 hypotheses in the beam was correct.

## 4.4 RESULTS

Table 2 reports the accuracy of our model for function integration and differential equations. For integration, the model achieves close to 100% performance on a held-out test set, even with greedy decoding (beam size 1). This performance is consistent over the three integration datasets (`FWD`, `BWD`, and `IBP`). Greedy decoding (beam size 1) does not work as well for differential equations. In particular, we observe an improvement in accuracy of almost 40% when using a large beam size of 50 for second order differential equations. Unlike in machine translation, where increasing the beam size does not necessarily increase the performance (Ott et al., 2018), we always observe significant improvements with wider beams. Typically, using a beam size of 50 provides an improvement of 8% accuracy compared to a beam size of 10. This makes sense, as increasing the beam size will provide more hypotheses, although a wider beam may displace a valid hypothesis to consider invalid ones with better log-probabilities.

|             | Integration (`FWD`) | Integration (`BWD`) | Integration (`IBP`) | ODE (order 1) | ODE (order 2) |
|-------------|---------------------|---------------------|---------------------|---------------|---------------|
| Beam size 1  | 93.6 | 98.4 | 96.8 | 77.6 | 43.0 |
| Beam size 10 | 95.6 | 99.4 | 99.2 | 90.5 | 73.0 |
| Beam size 50 | 96.2 | 99.7 | 99.5 | 94.0 | 81.2 |

Table 2: **Accuracy of our models on integration and differential equation solving.** Results are reported on a held out test set of 5000 equations. For differential equations, using beam search decoding significantly improves the accuracy of the model.

## 4.5 COMPARISON WITH MATHEMATICAL FRAMEWORKS

We compare our model with three popular mathematical frameworks: Mathematica (Wolfram-Research, 2019), Maple and Matlab (MathWorks, 2019)[1]. Prefix sequences in our test set are

---

[1]All experiments were run with Mathematica 12.0.0.0, Maple 2019 and Matlab R2019a.

converted back to their infix representations, and given as input to the computer algebra. For a specific input, the computer algebra either returns a solution, provides no solution (or a solution including integrals or special functions), or, in the case of Mathematica, times out after a preset delay. When Mathematica times out, we conclude that it is not able to compute a solution (although it might have found a solution given more time). For integration, we evaluate on the `BWD` test set. By construction, the `FWD` data only consists of integrals generated by computer algebra systems, which makes comparison uninteresting.

In Table 3, we present accuracy for our model with different beam sizes, and for Mathematica with a timeout delay of 30 seconds. Table 8 in the appendix provides detailed results for different values of timeout, and explains our choice of 30 seconds. In particular, we find that with 30 seconds, only 20% of failures are due to timeouts, and only 10% when the timeout is set to 3 minutes. Even with timeout limits, evaluation would take too long on our 5000 test equations, so we only evaluate on a smaller test subset of 500 equations, on which we also re-evaluate our model.

|  | Integration (`BWD`) | ODE (order 1) | ODE (order 2) |
|---|---|---|---|
| Mathematica (30s) | 84.0 | 77.2 | 61.6 |
| Matlab | 65.2 | - | - |
| Maple | 67.4 | - | - |
| Beam size 1 | 98.4 | 81.2 | 40.8 |
| Beam size 10 | 99.6 | 94.0 | 73.2 |
| Beam size 50 | 99.6 | 97.0 | 81.0 |

Table 3: **Comparison of our model with Mathematica, Maple and Matlab on a test set of 500 equations.** For Mathematica we report results by setting a timeout of 30 seconds per equation. On a given equation, our model typically finds the solution in less than a second.

On all tasks, we observe that our model significantly outperforms Mathematica. On function integration, our model obtains close to 100% accuracy, while Mathematica barely reaches 85%. On first order differential equations, Mathematica is on par with our model when it uses a beam size of 1, i.e. with greedy decoding. However, using a beam search of size 50 our model accuracy goes from 81.2% to 97.0%, largely surpassing Mathematica. Similar observations can be made for second order differential equations, where beam search is even more critical since the number of equivalent solutions is larger. On average, Matlab and Maple have slightly lower performance than Mathematica on the problems we tested.

Table 4 shows examples of functions that our model was able to solve, on which Mathematica and Matlab did not find a solution. The denominator of the function to integrate, $-16x^8 + 112x^7 - 204x^6 + 28x^5 - x^4 + 1$, can be rewritten as $1 - (4x^4 - 14x^3 + x^2)^2$. With the simplified input:

$$\frac{16x^3 - 42x^2 + 2x}{\left(1 - (4x^4 - 14x^3 + x^2)^2\right)^{1/2}}$$

integration becomes easier and Mathematica is able to find the solution.

| Equation | Solution |
|---|---|
| $y' = \dfrac{16x^3 - 42x^2 + 2x}{(-16x^8 + 112x^7 - 204x^6 + 28x^5 - x^4 + 1)^{1/2}}$ | $y = \sin^{-1}(4x^4 - 14x^3 + x^2)$ |
| $3xy\cos(x) - \sqrt{9x^2\sin(x)^2 + 1}\,y' + 3y\sin(x) = 0$ | $y = c\exp\left(\sinh^{-1}(3x\sin(x))\right)$ |
| $4x^4yy'' - 8x^4y'^2 - 8x^3yy' - 3x^3y'' - 8x^2y^2 - 6x^2y' - 3x^2y'' - 9xy' - 3y = 0$ | $y = \dfrac{c_1 + 3x + 3\log(x)}{x(c_2 + 4x)}$ |

Table 4: Examples of problems that our model is able to solve, on which Mathematica and Matlab were not able to find a solution. For each equation, our model finds a valid solution with greedy decoding.

### 4.6 EQUIVALENT SOLUTIONS

An interesting property of our model is that it is able to generate solutions that are exactly equivalent, but written in different ways. For instance, we consider the following first order differential equation, along with one of its solutions:

$$162x \log(x)y' + 2y^3 \log(x)^2 - 81y \log(x) + 81y = 0 \qquad y = \frac{9\sqrt{x}\sqrt{\frac{1}{\log(x)}}}{\sqrt{c + 2x}}$$

In Table 5, we report the top 10 hypotheses returned by our model for this equation. We observe that all generations are actually valid solutions, although they are expressed very differently. They are however not all equal: merging the square roots within the first and third equations would give the same expression except that the third one would contain a factor 2 in front of the constant $c$, but up to a change of variable, these two solutions are actually equivalent. The ability of the model to recover equivalent expressions, without having been trained to do so, is very intriguing.

| Hypothesis | Score | Hypothesis | Score |
|---|---|---|---|
| $\dfrac{9\sqrt{x}\sqrt{\frac{1}{\log(x)}}}{\sqrt{c + 2x}}$ | $-0.047$ | $\dfrac{9}{\sqrt{\frac{c\log(x)}{x} + 2\log(x)}}$ | $-0.124$ |
| $\dfrac{9\sqrt{x}}{\sqrt{c + 2x}\sqrt{\log(x)}}$ | $-0.056$ | $\dfrac{9\sqrt{x}}{\sqrt{c\log(x) + 2x\log(x)}}$ | $-0.139$ |
| $\dfrac{9\sqrt{2}\sqrt{x}\sqrt{\frac{1}{\log(x)}}}{2\sqrt{c + x}}$ | $-0.115$ | $\dfrac{9}{\sqrt{\frac{c}{x} + 2}\sqrt{\log(x)}}$ | $-0.144$ |
| $9\sqrt{x}\sqrt{\dfrac{1}{c\log(x) + 2x\log(x)}}$ | $-0.117$ | $9\sqrt{\dfrac{1}{\frac{c\log(x)}{x} + 2\log(x)}}$ | $-0.205$ |
| $\dfrac{9\sqrt{2}\sqrt{x}}{2\sqrt{c + x}\sqrt{\log(x)}}$ | $-0.124$ | $9\sqrt{x}\sqrt{\dfrac{1}{c\log(x) + 2x\log(x) + \log(x)}}$ | $-0.232$ |

Table 5: Top 10 generations of our model for the first order differential equation $162x \log(x)y' + 2y^3 \log(x)^2 - 81y \log(x) + 81y = 0$, generated with a beam search. All hypotheses are valid solutions, and are equivalent up to a change of the variable $c$. Scores are log-probabilities normalized by sequence lengths.

### 4.7 GENERALIZATION ACROSS GENERATORS

Models for integration achieve close to 100% performance on held-out test samples generated with the same method as their training data. In Table 6, we compare the accuracy on the `FWD`, `BWD` and `IBP` test sets for 4 models trained using different combinations of training data. When the test set is generated with the same generator as the training set, the model performs extremely well. For instance, the three models trained either on `BWD`, `BWD + IBP` or `BWD + IBP + FWD` achieve 99.7% accuracy on the `BWD` test set with a beam size of 50.

On the other hand, even with a beam size of 50, a `FWD`-trained model only achieves 17.2% accuracy on the `BWD` test set, and a `BWD`-trained model achieves 27.5% on the `FWD` test set. This results from the very different structure of the `FWD` and `BWD` data sets (cf. Table 1 and the discussion in Section E of the appendix). Overall, a model trained on `BWD` samples learns that integration tends to shorten expressions, a property that does not hold for `FWD` samples. Adding diversity to the training set improves the results. For instance, adding `IBP`-generated examples to the `BWD`-trained model raises the `FWD` test accuracy from 27.5% to 56.1%, and with additional `FWD` training data the model reaches 94.3% accuracy. Generalization is further discussed in Section E of the appendix.

| Training data | Forward (FWD) | | | Backward (BWD) | | | Integration by parts (IBP) | | |
|---|---|---|---|---|---|---|---|---|---|
| | Beam 1 | Beam 10 | Beam 50 | Beam 1 | Beam 10 | Beam 50 | Beam 1 | Beam 10 | Beam 50 |
| FWD | 93.6 | 95.6 | 96.2 | 10.9 | 13.9 | 17.2 | 85.6 | 86.8 | 88.9 |
| BWD | 18.9 | 24.6 | 27.5 | 98.4 | 99.4 | 99.7 | 42.9 | 54.6 | 59.2 |
| BWD + IBP | 41.6 | 54.9 | 56.1 | 98.2 | 99.4 | 99.7 | 96.8 | 99.2 | 99.5 |
| BWD + IBP + FWD | 89.1 | 93.4 | 94.3 | 98.1 | 99.3 | 99.7 | 97.2 | 99.4 | 99.7 |

Table 6: **Accuracy of our models on function integration.** We report the accuracy of our model on the three integration datasets: forward (FWD), backward (BWD), and integration by parts (IBP), for four models trained with different combinations of training data. We observe that a FWD-trained model performs poorly when it tries to integrate functions from the BWD dataset. Similarly, a BWD-trained model only obtain 27.5% accuracy on the FWD dataset, as it fails to integrate simple functions like $x^5 \sin(x)$. On the other hand, training on both the BWD + IBP datasets allows the model to reach up to 56.1% accuracy on FWD. Training on all datasets allows the model to perform well on the three distributions.

## 4.8 GENERALIZATION BEYOND THE GENERATOR - SYMPY

Our forward generator, FWD, generates a set of pairs $(f, F)$ of functions with their integrals. It relies on an external symbolic framework, SymPy (Meurer et al., 2017), to compute the integral of randomly generated functions. SymPy is not perfect, and fails to compute the integral of many integrable functions. In particular, we found that the accuracy of SymPy on the BWD test set is only 30%. Our FWD-trained model only obtains an accuracy of 17.2% on BWD. However, we observed that the FWD-trained model is sometimes able to compute the integral of functions that SymPy cannot compute. This means that by only training on functions that SymPy can integrate, the model was able to generalize to functions that SymPy cannot integrate. Table 7 presents examples of such functions with their integrals.

| | |
|---|---|
| $x^2 \left( \tan^2 (x) + 1 \right) + 2x \tan (x) + 1$ | $x^2 \tan (x) + x$ |
| $1 + \dfrac{2 \cos (2x)}{\sqrt{\sin^2 (2x) + 1}}$ | $x + \operatorname{asinh} (\sin (2x))$ |
| $\dfrac{x \tan (x) + \log (x \cos (x)) - 1}{\log (x \cos (x))^2}$ | $\dfrac{x}{\log (x \cos (x))}$ |
| $-\dfrac{2x \cos \left( \operatorname{asin}^2 (x) \right) \operatorname{asin} (x)}{\sqrt{1 - x^2} \sin^2 \left( \operatorname{asin}^2 (x) \right)} + \dfrac{1}{\sin \left( \operatorname{asin}^2 (x) \right)}$ | $\dfrac{x}{\sin \left( \operatorname{asin}^2 (x) \right)}$ |
| $\sqrt{x} + x \left( \dfrac{2x}{\sqrt{x^4 + 1}} + 1 + \dfrac{1}{2\sqrt{x}} \right) + x + \operatorname{asinh} \left( x^2 \right)$ | $x \left( \sqrt{x} + x + \operatorname{asinh} \left( x^2 \right) \right)$ |
| $-3 - \dfrac{3 \left( -3x^2 \sin \left( x^3 \right) + \frac{1}{2\sqrt{x}} \right)}{\sqrt{x} + \cos \left( x^3 \right)}}{\left( x + \log \left( \sqrt{x} + \cos \left( x^3 \right) \right) \right)^2}$ | $\dfrac{3}{x + \log \left( \sqrt{x} + \cos \left( x^3 \right) \right)}$ |
| $\dfrac{-2 \tan^2 (\log (\log (x))) - 2}{\log (x) \tan^2 (\log (\log (x)))} + \dfrac{2}{\tan (\log (\log (x)))}$ | $\dfrac{2x}{\tan (\log (\log (x)))}$ |

Table 7: **Examples of functions / integrals that the FWD-trained model can integrate, but not SymPy.** Although the FWD model was only trained on a subset of functions that SymPy can integrate, it learned to generalize to functions that SymPy cannot integrate.

## 5 RELATED WORK

Computers were used for symbolic mathematics since the late 1960s (Moses, 1974). Computer algebra systems (CAS), such as Matlab, Mathematica, Maple, PARI and SAGE, are used for a variety of mathematical tasks (Gathen & Gerhard, 2013). Modern methods for symbolic integration are based on Risch algorithm (Risch, 1970). Implementations can be found in Bronstein (2005) and Geddes et al. (1992). However, the complete description of the Risch algorithm takes more than 100 pages, and is not fully implemented in current mathematical framework.

Deep learning networks have been used to simplify treelike expressions. Zaremba et al. (2014) use recursive neural networks to simplify complex symbolic expressions. They use tree representations for expressions, but provide the model with problem related information: possible rules for simplification. The neural network is trained to select the best rule. Allamanis et al. (2017) propose a framework called *neural equivalence networks* to learn semantic representations of algebraic expressions. Typically, a model is trained to map different but equivalent expressions (like the 10 expressions proposed in Table 5) to the same representation. However, they only consider Boolean and polynomial expressions. More recently, Arabshahi et al. (2018a;b) used tree-structured neural networks to verify the correctness of given symbolic entities, and to predict missing entries in incomplete mathematical equations. They also showed that these networks could be used to predict whether an expression is a valid solution of a given differential equation.

Most attempts to use deep networks for mathematics have focused on arithmetic over integers (sometimes over polynomials with integer coefficients). For instance, Kaiser & Sutskever (2015) proposed the Neural-GPU architecture, and train networks to perform additions and multiplications of numbers given in their binary representations. They show that a model trained on numbers with up-to 20 bits can be applied to much larger numbers at test time, while preserving a perfect accuracy. Freivalds & Liepins (2017) proposed an improved version of the Neural-GPU by using hard non-linear activation functions, and a diagonal gating mechanism.

Saxton et al. (2019) use LSTMs (Hochreiter & Schmidhuber, 1997) and transformers on a wide range of problems, from arithmetic to simplification of formal expressions. However, they only consider polynomial functions, and the task of differentiation, which is significantly easier than integration. Trask et al. (2018) propose the Neural arithmetic logic units, a new module designed to learn systematic numerical computation, and that can be used within any neural network. Like Kaiser & Sutskever (2015), they show that at inference their model can extrapolate on numbers orders of magnitude larger than the ones seen during training.

## 6 CONCLUSION

In this paper, we show that standard seq2seq models can be applied to difficult tasks like function integration, or solving differential equations. We propose an approach to generate arbitrarily large datasets of equations, with their associated solutions. We show that a simple transformer model trained on these datasets can perform extremely well both at computing function integrals, and solving differential equations, outperforming state-of-the-art mathematical frameworks like Matlab or Mathematica that rely on a large number of algorithms and heuristics, and a complex implementation (Risch, 1970). Results also show that the model is able to write identical expressions in very different ways.

These results are surprising given the difficulty of neural models to perform simpler tasks like integer addition or multiplication. However, proposed hypotheses are sometimes incorrect, and considering multiple beam hypotheses is often necessary to obtain a valid solution. The validity of a solution itself is not provided by the model, but by an external symbolic framework (Meurer et al., 2017). These results suggest that in the future, standard mathematical frameworks may benefit from integrating neural components in their solvers.

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

## A    A SYNTAX FOR MATHEMATICAL EXPRESSIONS

We represent mathematical expressions as trees with operators as internal nodes, and numbers, constants or variables, as leaves. By enumerating nodes in prefix order, we transform trees into sequences suitable for seq2seq architectures.

For this representation to be efficient, we want expressions, trees and sequences to be in a one-to-one correspondence. Different expressions will always result in different trees and sequences, but for the reverse to hold, we need to take care of a few special cases.

First, expressions like sums and products may correspond to several trees. For instance, the expression $2 + 3 + 5$ can be represented as any one of those trees:

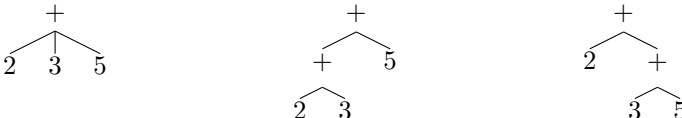

We will assume that all operators have at most two operands, and that, in case of doubt, they are associative to the right. $2 + 3 + 5$ would then correspond to the rightmost tree.

Second, the distinction between internal nodes (operators) and leaves (mathematical primitive objects) is somewhat arbitrary. For instance, the number $-2$ could be represented as a basic object, or as a unary minus operator applied to the number 2. Similarly, there are several ways to represent $\sqrt{5}$, $42x^5$, or the function $\log_{10}$. For simplicity, we only consider numbers, constants and variables as possible leaves, and avoid using a unary minus. In particular, expressions like $-x$ are represented as $-1 \times x$. Here are the trees for $-2$, $\sqrt{5}$, $42x^5$ and $-x$:

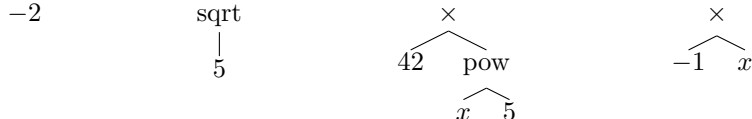

Integers are represented in positional notation, as a sign followed by a sequence of digits (from 0 to 9 in base 10). For instance, $2354$ and $-34$ are represented as $+2\ 3\ 5\ 4$ and $-\ 3\ 4$. For zero, a unique representation is chosen ($+0$ or $-0$).

## B    MATHEMATICAL DERIVATIONS OF THE PROBLEM SPACE SIZE

In this section, we investigate the size of the problem space by computing the number of expressions with $n$ internal nodes. We first deal with the simpler case where we only have binary operators ($p_1 = 0$), then consider trees and expressions composed of unary and binary operators. In each case, we calculate a generating function (Flajolet & Sedgewick, 2009; Wilf, 2005) from which we derive a closed formula or recurrence on the number of expressions, and an asymptotic expansion.

### B.1    BINARY TREES AND EXPRESSIONS

The main part of this derivation follows (Knuth, 1997) (pages 388-389).

**Generating function**    Let $b_n$ be the number of binary trees with $n$ internal nodes. We have $b_0 = 1$ and $b_1 = 1$. Any binary tree with $n$ internal nodes can be generated by concatenating a left and a right subtree with $k$ and $n - 1 - k$ internal nodes respectively. By summing over all possible values of $k$, we have that:

$$b_n = b_0 b_{n-1} + b_1 b_{n-2} + \cdots + b_{n-2} b_1 + b_{n-1} b_0$$

Let $B(z)$ be the generating function of $b_n$, $B(z) = b_0 + b_1 z + b_2 z^2 + b_3 z^3 + \dots$

$$B(z)^2 = b_0{}^2 + (b_0 b_1 + b_1 b_0)z + (b_0 b_2 + b_1 b_1 + b2b_0)z^2 + \ldots$$
$$= b_1 + b_2 z + b_3 z^2 + \ldots$$
$$= \frac{B(z) - b_0}{z}$$

So, $zB(z)^2 - B(z) + 1 = 0$. Solving for $B(z)$ gives:

$$B(z) = \frac{1 \pm \sqrt{1 - 4z}}{2z}$$

and since $B(0) = b_0 = 1$, we derive the generating function for sequence $b_n$

$$B(z) = \frac{1 - \sqrt{1 - 4z}}{2z}$$

We now derive a closed formula for $b_n$. By the binomial theorem,

$$B(z) = \frac{1}{2z}\left(1 - \sum_{k=0}^{\infty}\binom{1/2}{k}(-4z)^k\right)$$
$$= \frac{1}{2z}\left(1 + \sum_{k=0}^{\infty}\frac{1}{2k-1}\binom{2k}{k}z^k\right)$$
$$= \frac{1}{2z}\sum_{k=1}^{\infty}\frac{1}{2k-1}\binom{2k}{k}z^k$$
$$= \sum_{k=1}^{\infty}\frac{1}{2(2k-1)}\binom{2k}{k}z^{k-1}$$
$$= \sum_{k=0}^{\infty}\frac{1}{2(2k+1)}\binom{2k+2}{k+1}z^k$$
$$= \sum_{k=0}^{\infty}\frac{1}{k+1}\binom{2k}{k}z^k$$

Therefore

$$b_n = \frac{1}{n+1}\binom{2n}{n} = \frac{(2n)!}{(n+1)!n!}$$

These are the Catalan numbers, a closed formula for the number of binary trees with $n$ internal nodes. We now observe that a binary tree with $n$ internal nodes has exactly $n + 1$ leaves. Since each node in a binary tree can represent $p_2$ operators, and each leaf can take $L$ values, we have that a tree with $n$ nodes can take $p_2^n L^{n+1}$ possible combinations of operators and leaves. As a result, the number of binary expressions with $n$ operators is given by:

$$E_n = \frac{(2n)!}{(n+1)!n!}p_2^n L^{n+1}$$

**Asymptotic estimate**   To derive an asymptotic approximation of $b_n$, we apply the Stirling formula:

$$n! \approx \sqrt{2\pi n}\left(\frac{n}{e}\right)^n \quad \text{so} \quad \binom{2n}{n} \approx \frac{4^n}{\sqrt{\pi n}} \quad \text{and} \quad b_n \approx \frac{4^n}{n\sqrt{\pi n}}$$

Finally, we have the following formulas for the number of expressions with $n$ internal nodes:

$$E_n \approx \frac{1}{n\sqrt{\pi n}}(4p_2)^n L^{n+1}$$

### B.2 Unary-binary trees

**Generating function** Let $s_n$ be the number of unary-binary trees (i.e. trees where internal nodes can have one or two children) with $n$ internal nodes. We have $s_0 = 1$ and $s_1 = 2$ (the only internal node is either unary or binary).

Any tree with $n$ internal nodes is obtained either by adding a unary internal node at the root of a tree with $n - 1$ internal nodes, or by concatenating with a binary operator a left and a right subtree with $k$ and $n - 1 - k$ internal nodes respectively. Summing up as before, we have:

$$s_n = s_{n-1} + s_0 s_{n-1} + s_1 s_{n-2} + \cdots + s_{n-1} s_0$$

Let $S(z)$ be the generating function of the $s_n$. The above formula translates into

$$S(z)^2 = \frac{S(z) - s_0}{z} - S(z)$$

$$zS(z)^2 + (z - 1)S(z) + 1 = 0$$

solving and taking into account the fact that $S(0) = 1$, we obtain the generating function of the $s_n$

$$S(z) = \frac{1 - z - \sqrt{1 - 6z + z^2}}{2z}$$

The numbers $s_n$ generated by $S(z)$ are known as the Schroeder numbers (OEIS A006318) (Sloane, 1996). They appear in different combinatorial problems (Stanley, 2011). Notably, they correspond to the number of paths from $(0, 0)$ to $(n, n)$ of a $n \times n$ grid, moving north, east, or northeast, and never rising above the diagonal.

**Calculation** Schroeder numbers do not have a simple closed formula, but a recurrence allowing for their calculation can be derived from their generating function. Rewriting $S(z)$ as

$$2zS(z) + z - 1 = -\sqrt{1 - 6z + z^2}$$

and differentiating, we have

$$2zS'(z) + 2S(z) + 1 = \frac{3 - z}{\sqrt{1 - 6z + z^2}} = \frac{3 - z}{1 - 6z + z^2}(1 - z - 2zS(z))$$

$$2zS'(z) + 2S(z)\left(1 + \frac{3z - z^2}{1 - 6z + z^2}\right) = \frac{(3 - z)(1 - z)}{1 - 6z + z^2} - 1$$

$$2zS'(z) + 2S(z)\frac{1 - 3z}{1 - 6z + z^2} = \frac{2 + 2z}{1 - 6z + z^2}$$

$$z(1 - 6z + z^2)S'(z) + (1 - 3z)S(z) = 1 + z$$

Replacing $S(z)$ and $S'(z)$ with their n-th coefficient yields, for $n > 1$

$$ns_n - 6(n - 1)s_{n-1} + (n - 2)s_{n-2} + s_n - 3s_{n-1} = 0$$

$$(n + 1)s_n = 3(2n - 1)s_{n-1} - (n - 2)s_{n-2}$$

Together with $s_0 = 1$ and $s_1 = 2$, this allows for fast ($O(n)$) calculation of Schroeder numbers.

**Asymptotic estimate** To derive an asymptotic formula of $s_n$, we develop the generating function around its smallest singularity (Flajolet & Odlyzko, 1990), i.e. the radius of convergence of the power series. Since

$$1 - 6z + z^2 = \left(1 - (3 - \sqrt{8})z\right)\left(1 - (3 + \sqrt{8})z\right)$$

The smallest singular value is

$$r_1 = \frac{1}{(3 + \sqrt{8})}$$

and the asymptotic formula will have the exponential term

$$r_1^{-n} = (3 + \sqrt{8})^n = (1 + \sqrt{2})^{2n}$$

In a neighborhood of $r_1$, the generating function can be rewritten as

$$S(z) \approx (1 + \sqrt{2})\left(1 - 2^{1/4}\sqrt{1 - (3 + \sqrt{8})z}\right) + O(1 - (3 + \sqrt{8})z)^{3/2}$$

Since

$$[z_n]\sqrt{1 - az} \approx -\frac{a^n}{\sqrt{4\pi n^3}}$$

where $[z_n]F(z)$ denotes the n-th coefficient in the formal series of F, we have

$$s_n \approx \frac{(1 + \sqrt{2})(3 + \sqrt{8})^n}{2^{3/4}\sqrt{\pi n^3}} = \frac{(1 + \sqrt{2})^{2n+1}}{2^{3/4}\sqrt{\pi n^3}}$$

Comparing with the number of binary trees, we have

$$s_n \approx 1.44(1.46)^n b_n$$

### B.3 Unary-binary expressions

In the binary case, the number of expressions can be derived from the number of trees. This cannot be done in the unary-binary case, as the number of leaves in a tree with $n$ internal nodes depends on the number of binary operators ($n_2 + 1$).

**Generating function** The number of trees with $n$ internal nodes and $n_2$ binary operators can be derived from the following observation: any unary-binary tree with $n_2$ binary internal nodes can be generated from a binary tree by adding unary internal nodes. Each node in the binary tree can receive one or several unary parents.

Since the binary tree has $2n_2 + 1$ nodes and the number of unary internal nodes to be added is $n - n_2$, the number of unary-binary trees that can be created from a specific binary tree is the number of multisets with $2n_2 + 1$ elements on $n - n_2$ symbols, that is

$$\binom{n + n_2}{n - n_2} = \binom{n + n_2}{2n_2}$$

If $b_q$ denotes the q-th Catalan number, the number of trees with $n_2$ binary operators among $n$ is

$$\binom{n + n_2}{2n_2}b_{n_2}$$

Since such trees have $n_2 + 1$ leaves, with $L$ leaves, $p_2$ binary and $p_1$ unary operators to choose from, the number of expressions is

$$E(n, n_2) = \binom{n + n_2}{2n_2}b_{n_2}p_2^{n_2}p_1^{n-n_2}L^{n_2+1}$$

Summing over all values of $n_2$ (from 0 to $n$) yields the number of different expressions

$$E_n = \sum_{n_2=0}^{n}\binom{n + n_2}{2n_2}b_{n_2}p_2^{n_2}p_1^{n-n_2}L^{n_2+1}z^n$$

Let $E(z)$ be the corresponding generating function.

$$E(z) = \sum_{n=0}^{\infty}E_n z^n$$

$$= \sum_{n=0}^{\infty}\sum_{n_2=0}^{n}\binom{n + n_2}{2n_2}b_{n_2}p_2^{n_2}p_1^{n-n_2}L^{n_2+1}z^n$$

$$= L\sum_{n=0}^{\infty}\sum_{n_2=0}^{n}\binom{n + n_2}{2n_2}b_{n_2}\left(\frac{Lp_2}{p_1}\right)^{n_2}p_1^n z^n$$

$$= L\sum_{n=0}^{\infty}\sum_{n_2=0}^{\infty}\binom{n + n_2}{2n_2}b_{n_2}\left(\frac{Lp_2}{p_1}\right)^{n_2}(p_1 z)^n$$

since $\binom{n+n_2}{2n_2} = 0$ when $n > n_2$

$$E(z) = L \sum_{n_2=0}^{\infty} b_{n_2} \left(\frac{Lp_2}{p_1}\right)^{n_2} \sum_{n=0}^{\infty} \binom{n+n_2}{2n_2} (p_1 z)^n$$

$$= L \sum_{n_2=0}^{\infty} b_{n_2} \left(\frac{Lp_2}{p_1}\right)^{n_2} \sum_{n=0}^{\infty} \binom{n+2n_2}{2n_2} (p_1 z)^{n+n_2}$$

$$= L \sum_{n_2=0}^{\infty} b_{n_2} (Lp_2 z)^{n_2} \sum_{n=0}^{\infty} \binom{n+2n_2}{2n_2} (p_1 z)^n$$

applying the binomial formula

$$E(z) = L \sum_{n_2=0}^{\infty} b_{n_2} (Lp_2 z)^{n_2} \frac{1}{(1-p_1 z)^{2n_2+1}}$$

$$= \frac{L}{1-p_1 z} \sum_{n_2=0}^{\infty} b_{n_2} \left(\frac{Lp_2 z}{(1-p_1 z)^2}\right)^{n_2}$$

applying the generating function for binary trees

$$E(z) = \frac{L}{1-p_1 z} \left(\frac{1 - \sqrt{1 - 4\frac{Lp_2 z}{(1-p_1 z)^2}}}{2\frac{Lp_2 z}{(1-p_1 z)^2}}\right)$$

$$= \frac{1-p_1 z}{2p_2 z} \left(1 - \sqrt{1 - 4\frac{Lp_2 z}{(1-p_1 z)^2}}\right)$$

$$= \frac{1-p_1 z - \sqrt{(1-p_1 z)^2 - 4Lp_2 z}}{2p_2 z}$$

Reducing, we have

$$E(z) = \frac{1 - p_1 z - \sqrt{1 - 2(p_1 + 2Lp_2 k)z + p_1 z^2}}{2p_2 z}$$

**Calculation** As before, there is no closed simple formula for $E_n$, but we can derive a recurrence formula by differentiating the generating function, rewritten as

$$2p_2 z E(z) + p_1 z - 1 = -\sqrt{1 - 2(p_1 + 2p_2 L)z + p_1 z^2}$$

$$2p_2 z E'(z) + 2p_2 E(z) + p_1 = \frac{p_1 + 2p_2 L - p_1 z}{\sqrt{1 - 2(p_1 + 2p_2 L)z + p_1 z^2}}$$

$$2p_2 z E'(z) + 2p_2 E(z) + p_1 = \frac{(p_1 + 2p_2 L - p_1 z)(1 - p_1 z - 2p_2 z E(z))}{1 - 2(p_1 + 2p_2 L)z + p_1 z^2}$$

$$2p_2 z E'(z) + 2p_2 E(z) \left(1 + \frac{z(p_1 + 2p_2 L - p_1 z)}{1 - 2(p_1 + 2p_2 L)z + p_1 z^2}\right) = \frac{(p_1 + 2p_2 L - p_1 z)(1 - p_1 z)}{1 - 2(p_1 + 2p_2 L)z + p_1 z^2} - p_1$$

$$2p_2 z E'(z) + 2p_2 E(z) \left(\frac{1 - (p_1 + 2p_2 L)z}{1 - 2(p_1 + 2p_2 L)z + p_1 z^2}\right) = \frac{2p_2 L(1 + p_1 z) + p_1(p_1 - 1)z}{1 - 2(p_1 + 2p_2 L)z + p_1 z^2}$$

$$2p_2 z E'(z)(1 - 2(p_1 + 2p_2 L)z + p_1 z^2) + 2p_2 E(z)(1 - (p_1 + 2p_2 L)z) = (2p_2 L(1 + p_1 z) + p_1(p_1 - 1)z)$$

replacing $E(z)$ and $E'(z)$ with their coefficients

$$2p_2(nE_n - 2(p_1 + 2p_2 L)(n-1)E_{n-1} + p_1(n-2)E(n-2)) + 2p_2(E_n - (p_1 + 2p_2 L)E_{n-1}) = 0$$

$$(n+1)E_n - (p_1 + 2p_2 L)(2n-1)E_{n-1} + p_1(n-2)E_{n-2} = 0$$

$$(n+1)E_n = (p_1 + 2p_2 L)(2n-1)E_{n-1} - p_1(n-2)E_{n-2}$$

which together with

$$E_0 = L$$
$$E_1 = (p_1 + p_2 L)L$$

provides a formula for calculating $E_n$.

**Asymptotic estimate**   As before, approximations of $E_n$ for large $n$ can be found by developing $E(z)$ in the neighbourhood of the root with the smallest module of

$$1 - 2(p_1 + 2p_2 L)z + p_1 z^2$$

The roots are

$$r_1 = \frac{p_1}{p_1 + 2p_2 L - \sqrt{p_1^2 + 4p_2^2 L^2 + 4p_2 p_1 L - p_1}}$$

$$r_2 = \frac{p_1}{p_1 + 2p_2 L + \sqrt{p_1^2 + 4p_2^2 L^2 + 4p_2 p_1 L - p_1}}$$

both are positive and the smallest one is $r_2$

To alleviate notation, let

$$\delta = \sqrt{p_1^2 + 4p_2^2 L^2 + 4p_2 p_1 L - p_1}$$

$$r_2 = \frac{p_1}{p_1 + 2p_2 L + \delta}$$

developing $E(z)$ near $r_2$,

$$E(z) \approx \frac{1 - p_1 r_2 - \sqrt{1 - r_2 \left(\frac{p_1 + 2p_2 L - \delta}{p_1}\right)} \sqrt{1 - \frac{z}{r_2}}}{2p_2 r_2} + O(1 - \frac{z}{r_2})^{3/2}$$

$$E(z) \approx \frac{p_1 + 2p_2 L + \delta - p_1^2 - \sqrt{p_1 + 2p_2 L + \delta} \sqrt{2\delta} \sqrt{1 - \frac{z}{r_2}}}{2p_2 p_1} + O(1 - \frac{z}{r_2})^{3/2}$$

and therefore

$$E_n \approx \frac{\sqrt{\delta} r_2^{-n-\frac{1}{2}}}{2p_2 \sqrt{2\pi p_1 n^3}} = \frac{\sqrt{\delta}}{2p_2 \sqrt{2\pi n^3}} \frac{(p_1 + 2p_2 L + \delta)^{n+\frac{1}{2}}}{p_1^{n+1}}$$

## C   GENERATING RANDOM EXPRESSIONS

In this section we present algorithms to generate random expressions with $n$ internal nodes. We achieve this by generating random trees, and selecting randomly their nodes and leaves. We begin with the simpler binary case ($p_1 = 0$).

### C.1   BINARY TREES

To generate a random binary tree with $n$ internal nodes, we use the following one-pass procedure. Starting with an empty root node, we determine at each step the position of the next internal nodes among the empty nodes, and repeat until all internal nodes are allocated.

Start with an empty node, set $e = 1$;
**while** $n > 0$ **do**
    | Sample a position $k$ from $K(e, n)$;
    | Sample the $k$ next empty nodes as leaves;
    | Sample an operator, create two empty children;
    | Set $e = e - k + 1$ and $n = n - 1$;
**end**

**Algorithm 1:** Generate a random binary tree

We denote by $e$ the number of empty nodes, by $n > 0$ the number of operators yet to be generated, and by $K(e, n)$ the probability distribution of the position (0-indexed) of the next internal node to allocate.

To calculate $K(e, n)$, let us define $D(e, n)$, the number of different binary subtrees that can be generated from $e$ empty elements, with $n$ internal nodes to generate. We have

$$D(0, n) = 0$$
$$D(e, 0) = 1$$
$$D(e, n) = D(e - 1, n) + D(e + 1, n - 1)$$

The first equation states that no tree can be generated with zero empty node and $n > 0$ operators. The second equation says that if no operator is to be allocated, empty nodes must all be leaves and there is only one possible tree. The last equation states that if we have $e > 0$ empty nodes, the first one is either a leaf (and there are $D(e - 1, n)$ such trees) or an internal node ($D(e + 1, n - 1)$ trees). This allows us to compute $D(e, n)$ for all $e$ and $n$.

To calculate distribution $K(e, n)$, observe that among the $D(e, n)$ trees with $e$ empty nodes and $n$ operators, $D(e + 1, n - 1)$ have a binary node in their first position. Therefore

$$P(K(e, n) = 0) = \frac{D(e + 1, n - 1)}{D(e, n)}$$

Of the remaining $D(e - 1, n)$ trees, $D(e, n - 1)$ have a binary node in their first position (same argument for $e - 1$), that is

$$P(K(e, n) = 1) = \frac{D(e, n - 1)}{D(e, n)}$$

By induction over $k$, we have the general formula

$$P\big(K(e, n) = k\big) = \frac{D(e - k + 1, n - 1)}{D(e, n)}$$

## C.2 UNARY-BINARY TREES

In the general case, internal nodes can be of two types: unary or binary. We adapt the previous algorithm by considering the two-dimensional probability distribution $L(e, n)$ of position (0-indexed) and arity of the next internal node (i.e. $P(L(e, n) = (k, a)$ is the probability that the next internal node is in position $k$ and has arity $a$).

Start with an empty node, set $e = 1$;
**while** $n > 0$ **do**
    Sample a position $k$ and arity $a$ from $L(e, n)$ (if $a = 1$ the next internal node is unary);
    Sample the $k$ next empty nodes as leaves;
    **if** $a = 1$ **then**
        Sample a unary operator;
        Create one empty child;
        Set $e = e - k$;
    **end**
    **else**
        Sample a binary operator;
        Create two empty children;
        Set $e = e - k + 1$;
    **end**
    Set $n = n - 1$;
**end**

**Algorithm 2:** Generate a random unary-binary tree

To compute $L(e, n)$, we derive $D(e, n)$, the number of subtrees with $n$ internal nodes that can be generated from $e$ empty nodes. We have, for all $n > 0$ and $e$:

$$D(0, n) = 0$$
$$D(e, 0) = 1$$
$$D(e, n) = D(e - 1, n) + D(e, n - 1) + D(e + 1, n - 1)$$

The first equation states that no tree can be generated with zero empty node and $n > 0$ operators. The second says that if no operator is to be allocated, empty nodes must all be leaves and there is only one possible tree. The third equation states that with $e > 0$ empty nodes, the first one will either be a leaf ($D(e - 1, n)$ possible trees), a unary operator ($D(e, n - 1)$ trees), or a binary operator ($D(e + 1, n - 1)$ trees).

To derive $L(e, n)$, we observe that among the $D(e, n)$ subtrees with $e$ empty nodes and $n$ internal nodes to be generated, $D(e, n - 1)$ have a unary operator in position zero, and $D(e + 1, n - 1)$ have a binary operator in position zero. As a result, we have

$$P\big(L(e, n) = (0, 1)\big) = \frac{D(e, n - 1)}{D(e, n)} \quad \text{and} \quad P\big(L(e, n) = (0, 2)\big) = \frac{D(e + 1, n - 1)}{D(e, n)}$$

As in the binary case, we can generalize these probabilities to all positions $k$ in $\{0 \ldots e - 1\}$

$$P\big(L(e, n) = (k, 1)\big) = \frac{D(e - k, n - 1)}{D(e, n)} \quad \text{and} \quad P\big(L(e, n) = (k, 2)\big) = \frac{D(e - k + 1, n - 1)}{D(e, n)}$$

### C.3    SAMPLING EXPRESSIONS

To generate expressions, we sample random trees (binary, or unary binary), that we "decorate" by randomly selecting their internal nodes and leaves from a list of possible operators or mathematical entities (integers, variables, constants).

Nodes and leaves can be selected uniformly, or according to a prior probability. For instance, integers between $-a$ and $a$ could be sampled so that small absolute values are more frequent than large ones. For operators, addition and multiplication could be more common than substraction and division.

If all $L$ leaves, $p_1$ and $p_2$ operators are equiprobable, an alternative approach to generation can be defined by computing $D(e, n)$ as

$$D(0, n) = 0$$
$$D(e, 0) = L^e$$
$$D(e, n) = LD(e - 1, n) + p_1 D(e, n - 1) + p_2 D(e + 1, n - 1)$$

and normalizing the probabilities $P(L(e, n))$ as

$$P\big(L(e, n) = (k, 1)\big) = \frac{L^e D(e - k, n - 1)}{D(e, n)} \quad \text{and} \quad P\big(L(e, n) = (k, 2)\big) = \frac{L^e D(e - k + 1, n - 1)}{D(e, n)}$$

Samples then become dependent on the number of possible leaves and operators.

## D    IMPACT OF TIMEOUT ON MATHEMATICA

In the case of Mathematica, we use function DSolve to solve differential equations, and function Integrate to integrate functions. Since computations can take a long time, we set a finite timeout to limit the time spent on each equation. Table 8 shows the impact of the timeout value on the accuracy with Mathematica. Increasing the timeout delay naturally improves the accuracy. With a timeout of 30 seconds, Mathematica times out on 20% of unsolved equations. With a limit of 3 minutes, timeouts represent about 10% of failed equations. This indicates that even in the ideal scenario where Mathematica would succeed on all equations where it times out, the accuracy would not exceed 86.2%.

| Timeout (s) | Success | Failure | Timeout |
|---|---|---|---|
| 5 | 77.8 | 9.8 | 12.4 |
| 10 | 82.2 | 11.6 | 6.2 |
| 30 | 84.0 | 12.8 | 3.2 |
| 60 | 84.4 | 13.4 | 2.2 |
| 180 | 84.6 | 13.8 | 1.6 |

Table 8: **Accuracy of Mathematica on 500 functions to integrate, for different timeout values.** As the timeout delay increases, the percentage of failures due to timeouts decreases. With a limit of 3 minutes, timeouts only represent 10% of failures. As a result, the accuracy without timeout would not exceed 86.2%.

# E GENERALIZATION ACROSS GENERATORS

On the integration problem, we achieve (c.f. Table 6) near perfect performance when the training and test data are generated by the same method (either FWD, BWD, or IBP). Given the relatively small size of the training set ($4.10^7$ examples), the model cannot overfit to the entire problem space ($10^{34}$ possible expressions). This shows that:

- Our model generalizes well to functions created by the training generator.
- This property holds for the three considered generators, FWD, BWD, and IBP.

Table 6 also measures the ability of our model to generalize across generators. A FWD-trained model achieves a low performance (17.2% with beam 50) on a BWD-generated test set. A BWD-trained model does a little better on the FWD test set (27.5%), but accuracy remains low. On the other hand, FWD-trained models achieve very good accuracy over an IBP-generated test set (88.9%), and BWD-trained models stand in the middle (59.2%).

Figure 2 provides an explanation for these results. The input/output pairs produced by FWD and BWD have very different distributions: integration tends to shorten BWD generated expressions, and to expand FWD generated expressions. As a result, a model trained on BWD generated data will learn this shortening feature of integration, which will prove wrong on a FWD test set. Similar problems will happen on a FWD trained model with a BWD test set. Since IBP keeps average expression lengths unchanged, BWD and FWD-trained models will generalize better to IBP test sets (and be more accurate on FWD-trained models, since their input length distributions are closer).

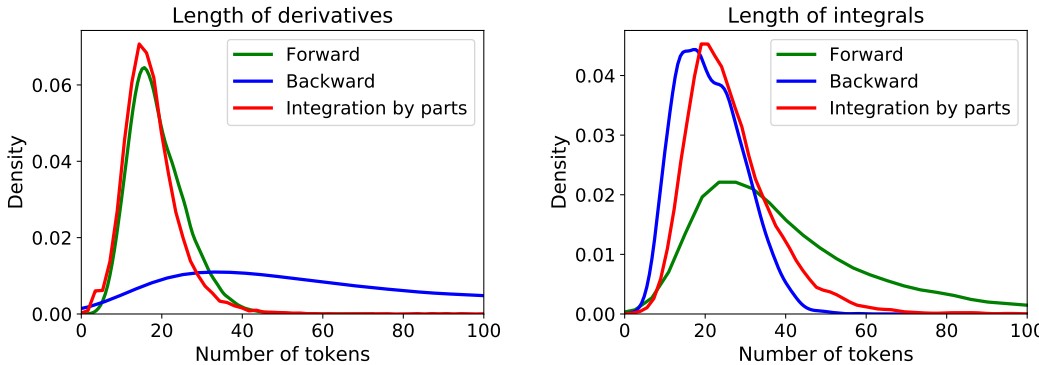

Figure 2: **Distribution of input and output lengths for different integration datasets.** The FWD generator produces short problems with long solutions. Conversely, the BWD generator creates long problems, with short solutions. The IBP approach stands in the middle, and generates short problems with short solutions.

This suggests that what looks at first glance like a generalization problem (bad accuracy of BWD-trained models on FWD generated sets, and the converse) is in fact a consequence of data generation. BWD and FWD methods generate training sets with specific properties, that our model will learn. But this can be addressed by adding IBP or FWD data to the BWD dataset, as shown in the two last lines of Table 6. In practice, a better approach could be implemented with self-supervised learning, where new training examples are generated by the model itself.

| Functions and their primitives generated with the forward approach (FWD) | |
|---|---|
| $\cos^{-1}(x)$ | $x\cos^{-1}(x) - \sqrt{1-x^2}$ |
| $x\left(2x + \cos\left(2x\right)\right)$ | $\dfrac{2x^3}{3} + \dfrac{x\sin\left(2x\right)}{2} + \dfrac{\cos\left(2x\right)}{4}$ |
| $\dfrac{x\left(x+4\right)}{x+2}$ | $\dfrac{x^2}{2} + 2x - 4\log\left(x+2\right)$ |
| $\dfrac{\cos\left(2x\right)}{\sin\left(x\right)}$ | $\dfrac{\log\left(\cos\left(x\right)-1\right)}{2} - \dfrac{\log\left(\cos\left(x\right)+1\right)}{2} + 2\cos\left(x\right)$ |
| $3x^2\sinh^{-1}\left(2x\right)$ | $x^3\sinh^{-1}\left(2x\right) - \dfrac{x^2\sqrt{4x^2+1}}{6} + \dfrac{\sqrt{4x^2+1}}{12}$ |
| $x^3\log\left(x^2\right)^4$ | $\dfrac{x^4\log\left(x^2\right)^4}{4} - \dfrac{x^4\log\left(x^2\right)^3}{2} + \dfrac{3x^4\log\left(x^2\right)^2}{4} - \dfrac{3x^4\log\left(x^2\right)}{4} + \dfrac{3x^4}{8}$ |

| Functions and their primitives generated with the backward approach (BWD) | |
|---|---|
| $\cos\left(x\right) + \tan^2\left(x\right) + 2$ | $x + \sin\left(x\right) + \tan\left(x\right)$ |
| $\dfrac{1}{x^2\sqrt{x-1}\sqrt{x+1}}$ | $\dfrac{\sqrt{x-1}\sqrt{x+1}}{x}$ |
| $\left(\dfrac{2x}{\cos^2\left(x\right)} + \tan\left(x\right)\right)\tan\left(x\right)$ | $x\tan^2\left(x\right)$ |
| $\dfrac{x\tan\left(\frac{e^x}{x}\right) + \frac{(x-1)e^x}{\cos^2\left(\frac{e^x}{x}\right)}}{x}$ | $x\tan\left(\dfrac{e^x}{x}\right)$ |
| $1 + \dfrac{1}{\log\left(\log\left(x\right)\right)} - \dfrac{1}{\log\left(x\right)\log\left(\log\left(x\right)\right)^2}$ | $x + \dfrac{x}{\log\left(\log\left(x\right)\right)}$ |
| $-2x^2\sin\left(x^2\right)\tan\left(x\right) + x\left(\tan^2\left(x\right)+1\right)\cos\left(x^2\right) + \cos\left(x^2\right)\tan\left(x\right)$ | $x\cos\left(x^2\right)\tan\left(x\right)$ |

| Functions and their primitives generated with the integration by parts approach (IBP) | |
|---|---|
| $x\left(x + \log\left(x\right)\right)$ | $\dfrac{x^2\left(4x + 6\log\left(x\right) - 3\right)}{12}$ |
| $\dfrac{x}{\left(x+3\right)^2}$ | $\dfrac{-x + \left(x+3\right)\log\left(x+3\right)}{x+3}$ |
| $\dfrac{x+\sqrt{2}}{\cos^2\left(x\right)}$ | $\left(x+\sqrt{2}\right)\tan\left(x\right) + \log\left(\cos\left(x\right)\right)$ |
| $x\left(2x+5\right)\left(3x + 2\log\left(x\right) + 1\right)$ | $\dfrac{x^2\left(27x^2 + 24x\log\left(x\right) + 94x + 90\log\left(x\right)\right)}{18}$ |
| $\dfrac{\left(x - \frac{2x}{\sin^2\left(x\right)} + \frac{1}{\tan\left(x\right)}\right)\log\left(x\right)}{\sin\left(x\right)}$ | $\dfrac{x\log\left(x\right) + \tan\left(x\right)}{\sin\left(x\right)\tan\left(x\right)}$ |
| $x^3\sinh\left(x\right)$ | $x^3\cosh\left(x\right) - 3x^2\sinh\left(x\right) + 6x\cosh\left(x\right) - 6\sinh\left(x\right)$ |

Table 9: **Examples of functions with their integrals, generated by our FWD, BWD and IBP approaches.** We observe that the FWD and IBP approaches tend to generate short functions, with long integrals, while the BWD approach generates short functions with long derivatives.

