# OpenReview forum: "Deep Learning For Symbolic Mathematics"
_ICLR.cc/2020/Conference — Accept (Spotlight)_

### Official Review · AnonReviewer1 · 2019-10-17
**Official Blind Review #1**

**Rating:** 6

**Review:**

It is rather interesting for a humble academic to review this paper. It already has a discussion, which I find very valuable, and many tweets and social media exposure and endorsements. It is onerous to review in this setting.

The paper makes a valuable contribution. The adversarial discussions in this website and the unhelpful hype can in this case be addressed to some extent by the authors. I will start with discussing this. Clearly, the title is too broad. This is not deep learning for symbolic mathematics. In no way does this paper address the essence of what is understood by "symbolic mathematics". What the authors address is mapping sequences of discrete quantities to other sequences of discrete quantities. The sequences in this paper correspond to function-integral i/o sequences, and 1st/2nd ODEs-function i/o sequences. I will leave it to the authors to come up with a more informative title, but something like deep learning or transformers for symbolic (1d) integration and simple ODEs with be far more accurate.

To hammer this point, note that Section 3 discusses removing "invalid" expressions: log(0) or sqrt(-2). However, it is the manipulation of infinity and imaginary numbers that could be considered to be one of the greatest achievements of symbolic mathematics over the last couple of hundred years. It is reasonable to expect neural nets to do this one day, because humans can, but this should come with results. It's too early to make the claim in the paper title.

Sentences such as "This suggest (sic) that some deeper understanding of mathematics has been achieved by the model." and "These results are surprising given the incapacity of neural models to perform simpler tasks ..." are speculative, potentially inaccurate and likely to increase hype. This hype is not needed.

Hype and over-claiming aside, I did enjoy reading this paper. The public commenters have already asked important questions about methodology and related work on neural programming that the authors have addressed in comments. I look forward to these being incorporated in the revised pdf.

A big part of the paper is about generating the datasets, and I therefore sympathise with the comment about requesting either a dataset release or the generating code. I see no obvious ethical concerns in this case, and the authors have already kindly offered to do this. This is a commendable and important service to our community and for this alone I would be inclined to vote for acceptance at ICLR.

The paper is clear and well written. However (i) it would be good to show several examples of input and output sequences (as done already in this website) and (ii) the Experiments section needs work. I'll expand on this next.

The seq2seq transformer with 8 heads, 6 layers and dimensionality 512 is a sensible choice. The authors should however explain why they expect this architecture to be able to map the sequences they adopt. That is, it is well known that a deep neural network is just a skeleton for an algorithm. By estimating the parameters, we are coming up with (fitting) the algorithm for the given datasets. What is the resulting algorithm? Why are 6 layers enough? Here some visualization would be helpful. See for example https://arxiv.org/pdf/1904.02679.pdf and https://arxiv.org/pdf/1906.04341.pdf For greater understanding of the problem, it may be useful to also try sparse transformers eg https://arxiv.org/abs/1805.08241

Beam search is a crucial component of the current solution. However, the authors simply cite Koehn 2004 for this. First, that work used language models to compute probabilities for beam search. I assume no language models are used in this case. What I'm getting to is that there are not enough details about the beam search in this paper. The authors should include pseudocode for the beam search and give a few examples. The paper (even better thesis) of Koehn is a good template for what should be included. This is important and should be explained.

For Mathematica, it would be useful to state it does other things and has not been optimized for the two tasks addressed in this paper only. It would also be useful, now that you have more time, to run it for a week or two and get answers not only for 30s but also for 60s. How often does it take longer than 30s? How do you score it then?

Please do include train and test curves. This would be helpful too. I will of course consider revising my score once the paper is updated.

Thanks for constructing this dataset and writing this paper. It is very interesting and promising.






**Experience Assessment:**

I have published in this field for several years.

**Review Assessment: Checking Correctness Of Derivations And Theory:**

I assessed the sensibility of the derivations and theory.

**Review Assessment: Checking Correctness Of Experiments:**

I assessed the sensibility of the experiments.

**Review Assessment: Thoroughness In Paper Reading:**

I read the paper at least twice and used my best judgement in assessing the paper.

---

> ### Author Response · Authors · 2019-11-09
> **Response to review #1 (1/2)**
>
> Thank you very much for your review and comments. We address your questions in order.
>
> PART 1/2
>
> ===== Hype / Overclaiming
>
> We had no control over the discussions on the Internet prior to the review, and took no part in them, nor did we encourage them by communicating on our work or publishing on arXiv before review. This is a side effect of the open review process, together with the very interesting adversarial discussions we just had.
>
> In the paper, we tried to be prudent and not overclaim, by explaining that we work with a dataset generated by our model and use standard differential equation solvers that may work better on different sets of equations. We also mention, at the end of paragraph 4.5 ”When comparing with Matlab and Mathematica, we work from a data set generated for our model and use their standard differential equation solvers. Different sets of equations, and advanced techniques for solving them (e.g. transforming them before introducing them in the solver) would probably result in smaller performance gaps.”
>
> ===== On sqrt(-2) and log(0), and the “cleaning” of some formulas
>
> The main reason why we eliminated such constants (and very large values such as exp(exp(exp(5))) is that they made life difficult for SymPy and NumPy, which we use to test and verify our results. They tended to cause unwanted (and sometimes very difficult to catch) exceptions, and even server crashes. Since our model works on symbols; and does not care for actual numeric values, these constants (as opposed to functions of variable x) had no impact on actual integration or equation solving, they could have been replaced by anything.
>
> Operating in the complex domain is also possible. We took the decision to discard complex equations arbitrarily, but we could easily add them back.
>
> However, on a deeper level, and in the specific case of symbolic integration, we do not think that adding infinity or operating in the complex domain would be an improvement. The objective of symbolic integration consists in finding a solution to an indefinite integral without adding new symbols, and in the smallest possible algebraic extension of the original field (here, an extension of Q since our constants are integers). We believe this is true for other tasks of symbolic mathematics.
>
> ===== On the two examples you provide
>
> "These results are surprising given the incapacity of neural models to perform simpler tasks like addition and multiplication"
> -> The difficulty to perform such calculations with neural networks is documented (see the reference in paragraph 2 of our introduction). We actually tested transformers on such problems (this was the original objective of our project), and were surprised to find that integration, a much more difficult task from a human point of view, seemed much easier for our model. We will clarify this.
>
> "This suggest (sic) that some deeper understanding of mathematics has been achieved by the model."
> -> We removed this sentence from the paper, but we consider that recovering equivalent expressions (i.e. alternative solutions of the problems) through beam search, is a very important finding.  As shown in Table 4 (Table 6 in the updated version of the paper), the model consistently recovers correct solutions that have very different representations. This is very surprising, and does suggest something important is at work. We have no explanation to offer so far, but we believe it is a very important observation.
>
> ===== Code / datasets
>
> Yes, as promised, we will make our code and datasets public after the review process.
>
> ===== Network architecture
>
> We decided to consider the same transformer configuration as Vaswani et al., i.e. 6 layers and a dimensionality of 512, with 8 heads. We tried to increase the number of layers, the number of heads, dimensionality, but did not observe significant improvements with larger models. On the other hand, we found that very small models (c.f. our response to Forough) still perform well on function integration, even when they are only composed of 2 layers of dimension 128. Our observation was that transformers perform well on the considered tasks, and are also very robust to the choice of hyper-parameters, unlike what people observed in machine translation. Machine translation systems typically benefit from advanced learning rate schedulers (either with linear or cosine decay, with many hyper-parameters). These schedulers did not bring any improvements in our case, and we simply use a constant learning rate of 10^(-4).

---

> > ### Author Response · Authors · 2019-11-09
> > **Response to review #1 (2/2)**
> >
> > ===== Visualization and sparse transformers
> >
> > We agree that visualizing the attention is very interesting, and could give insights on the way the model is actually operating. We tried to use the “bertviz” library to see whether some attention heads focus on specific sub-expressions in input equations. Unfortunately, we did not see any specific patterns in our visualizations. We also quickly tried something in the spirit to Malaviya et al. to constrain the attention of our model, hoping that visualization would be easier. Unlike them, we used a naive approach where we simply set to 0 the attention scores that are not in the top-k highest scores. We found that this constraint hurts the performance of the model for small values of k, and does not make visualization much easier because of the skip-connections in the transformer. We did not have time to investigate the visualization further, but will definitely consider it in future work.
> >
> > ===== Beam search
> >
> > The beam search procedure we use is the one described in Sutskever et al, 2014, which we now cite along Koehn 2004. We added another paragraph at the end of the Evaluation section to clarify how we use the beam search.
> >
> > ===== Mathematica comparison / timeouts
> >
> > After submission, we conducted more precise tests on Mathematica, over the same test set, and with the same trained model.
> >
> > For a given timeout delay, there are three possible outcomes:
> > 1- Mathematica finds a solution before it times out
> > 2- Mathematica times out without a solution
> > 3- Mathematica returns without a solution before time out (either by returning the input, or a solution including an indefinite integral)
> >
> > In the submission, we considered 2 and 3 as failures. In the experiments, we used one of several ways to compute integrals in Mathematica: function DSolve, which can be used both for integration and differential equations. Upon further investigation, we noticed that function Integrate runs faster, and therefore achieves better results for the same timeout value. We updated the scores of Mathematica in the paper, and added a table in the appendix (Table 7, Section D) with the percentage of outcomes (success, timeout, failure) for different timeout delays, computed using the faster Integrate.
> >
> > As the timeout delay increases, timeouts are less frequent, and a bound on infinite time success rate can be calculated. This suggests a success rate between 85% and 86%. This table also justifies 30s as a practical time out value. This reduces the gap between Mathematica and our model (as we had suggested in our paper), but a significant difference remains and this does not change the conclusions.
> >
> > ===== Training curves
> > We agree that training curves would be helpful and interesting. We will add some in the next revised version of the paper. Thank you for your suggestion.

---

### Official Review · AnonReviewer3 · 2019-10-20
**Official Blind Review #3**

**Rating:** 8

**Review:**

The authors use a Transformer neural network, originally architected for the purpose of language translation, to solve nontrivial mathematical equations, specifically integrals, first-order differential equations, and second-order differential equations. They also developed rigorous methods for sampling from a large space of relevant equations, which is critical for assembling the type of dataset needed for training such a data-intensive model.

Both the philosophical question posed by the paper (i.e. can neural networks designed for natural language sequence-to-sequence mappings be meaningfully applied to symbolic mathematics) and the resulting answer (i.e. yes, and such a neural network outperforms SOTA commercially-available systems) are interested in their own right, and together make a strong case for paper acceptance.

Details appearing in the OpenReview comments which should be explicitly specified in the paper before publication:
1) How large was the generated training set (40M), and how does this compare to the space of all equations under consideration (1e34).
2) The authors employ beam search in a non-standard manner, where they check for appearance of the equation solution among all of the generated candidates, rather than selecting the top-1. The fact that the reported accuracy with width-10 and width-50 beam searches are in effect measuring top-10 and top-50 accuracy should be clearly stated.


**Experience Assessment:**

I have read many papers in this area.

**Review Assessment: Checking Correctness Of Derivations And Theory:**

I did not assess the derivations or theory.

**Review Assessment: Checking Correctness Of Experiments:**

I carefully checked the experiments.

**Review Assessment: Thoroughness In Paper Reading:**

I read the paper at least twice and used my best judgement in assessing the paper.

---

> ### Author Response · Authors · 2019-11-09
> **Response to review #3**
>
> Thank you very much for your review and your comments. We address them in the updated version of the paper.
>
> In particular, we added a new table (Table 1 of the updated version) with statistics about the considered training sets, and the length of expressions. We also added a figure (Figure 1) that represents the number of trees and expressions for different numbers of operators and leaves.
>
> At the end of Section 4.3, we clarified our use of beam search, and explained how it differs from what people usually do in machine translation (i.e. only returning the hypothesis of the beam with the highest score).

---

### Official Review · AnonReviewer2 · 2019-10-24
**Official Blind Review #2**

**Rating:** 8

**Review:**

In this paper, the authors propose a method for generating two types of symbolic mathematics problems, integration and differential equations, and their solutions. The purpose of the method is to generate datasets for training transformer neural networks that solve integration and differential-equation problems. The authors note that while solving these problems is very difficult, generating solutions first and corresponding problems next automatically is feasible, and their method realizes this observation. The authors report that transformer networks trained on the synthetically generated solution-problem pairs outperform existing symbolic solvers for integration and differential equation.

Here are the reasons that I like the paper. The observation that solving a symbolic mathematics problem is often a pattern matching process is interesting. It is surprising to know that a transformer network designed to translated the generating problem-solution pairs backward (from problem to solution) works better than the solvers in Mathematica and Matlab. Also, I like nice cute tricks used in the authors' method for generating solution-problem pairs, such as the syntactic condition on a possible position of some constant. The paper is overall clearly written.

I presume that when the authors compare their learned solvers with Mathematica and Matlab, they used a dataset generated by their method. I feel that this comparison is somewhat unfair, although it still impresses me that even for this dataset, the authors' solvers beat Mathematica and Matlab. I suggest to try at least one more experiment on a dataset not generated by the authors' method (integration and differential equation problems from math textbooks or other sources) if possible.

* p3: Why is it important to have a generator that produces the four expression trees in p3 with equal or almost equal probabilities? Do you have any semi-formal or informal justification that the distribution of such a generator better matches the kind of expressions arising in the real world?

* p4: f(x)/x)) ===> f(x)/x)

* "If this equation can be solved in c1", p5: How realistic is this assumption?

* p5: 1/2 e^x(...) ===> 0 = 1/2 e^x(...)

* p5: If you have a thought or an observation on the impact of each of the data-cleaning steps in Section 3.4, I suggest you to share this in the paper.

* p6: Why did you remove expressions with more than 512 tokens?

* p6: compare to ===> compared to

* p7: Would you put the reminder of the size of the training set in Section 4.4? It only mentions that of the test set currently.

* p8: 1-(4x^2 ===> (1-(4x^2

**Experience Assessment:**

I have published one or two papers in this area.

**Review Assessment: Checking Correctness Of Derivations And Theory:**

I assessed the sensibility of the derivations and theory.

**Review Assessment: Checking Correctness Of Experiments:**

I assessed the sensibility of the experiments.

**Review Assessment: Thoroughness In Paper Reading:**

I read the paper thoroughly.

---

> ### Author Response · Authors · 2019-11-09
> **Response to review #2**
>
> Thank you very much for your review and comments. We address them in turn:
>
> ==== “I presume that when the authors compare their learned solvers with Mathematica and Matlab, they used a dataset generated by their method.”
>
> This is correct: we test our model and Mathematica on a held out sample from the generated sample (and mention at the end of paragraph 4.5 that this creates a favorable situation for our model).
>
> Since the submission, we tried to experiment on integration for samples generated with different methods. More precisely, we generated “forward” samples of random functions that SymPy knows how to integrate. This gives a good approximation of what Computer Algebras are good for. Examination of the samples shows that in backward samples, derivatives tend to be longer than primitives, whereas the opposite holds for forward samples. Unsurprisingly, a model trained on backward samples performs poorly on forward examples. But a forward-trained model achieves the same performance on forward data as a backward-trained model on backward data: this suggests that the performance is linked to data generation, and we actually observe that a model trained on the combination of backward and forward data achieves a good performance on all samples. These new results are in the updated version of the paper.
>
> ===== p3: Why is it important to have a generator that produces the four expression trees in p3 with equal or almost equal probabilities? Do you have any semi-formal or informal justification that the distribution of such a generator better matches the kind of expressions arising in the real world?
>
> We have no idea of the actual distribution of expressions “in the wild” (provided this has a meaning). Since we have no reason to consider an expression more relevant than another, we decided to sample all of them with the same probability. Since there is a one to one mapping from expressions to decorated trees (thanks to the prefix notation), we want to sample them uniformly, which means that all trees have to be sampled with the same probability.
>
> ===== "If this equation can be solved in c1", p5: How realistic is this assumption?
>
> Formally, the function F(x,y,c1) is the equation of the level curves of the function f, which we originally generated. The equation dF/dx = 0 corresponds to the gradient of F along x. Solving this equation in c1 amounts to finding an equation of the level curves of the gradient. In practice, we found that we can solve in c1 about 50% of the time. If we cannot, we simply discard the initial expression.
>
> ===== p5: If you have a thought or an observation on the impact of each of the data-cleaning steps in Section 3.4, I suggest you to share this in the paper.
>
> Equation simplification, like the use of small integer coefficients in expressions, limits the need for our model to carry out (and learn) arithmetic simplification in addition to the main task (integration of equation solving). This will reduce, or rather, bias, the generated expressions, by reducing the number of constants (i.e. leaves different from ‘x’ in the expression tree), and eliminating certain sequences of operators (exp(log()), sin(arcsin()), and so on. We consider it as a way to improve learning by focusing on the task at hand.
> Coefficient simplification is a trick of our method to generate differential equations. This step makes the elimination of constants c1 and c2 easier, but the generated equations and solutions remain the same.
> Invalid expression removal allows us to avoid exceptions when evaluating the functions. Since they only concern constants, they have very little impact on the problem. An alternative would be to replace the invalid sub-expressions by valid ones (see also our reply to reviewer 1 on this point).
>
> ===== p6: Why did you remove expressions with more than 512 tokens?
>
> We found that with very large expressions, the transformer model is subject to out of memory errors, which requires to use a smaller batch size at training time. To keep a large batch size (and to make training faster), we set this limit of 512 tokens. Overall, this is only discards a tiny fraction of the generated expressions.
>
> ==== p7: Would you put the reminder of the size of the training set in Section 4.4? It only mentions that of the test set currently.
>
> Yes, we added a new Table (table 1 in the updated version of the paper) with statistics about our datasets. Thank you for the suggestion.

---

> > ### Comment · AnonReviewer2 · 2019-11-14
> > **Thanks!**
> >
> > Thank you for your detailed response. It helped me to understand the paper better.

---

### Public Comment · ~A_B_C2 · 2019-09-26
**Error in figure for expressions as trees**

The third tree on the furthest right is incorrect and does not match the equation presented in the text

---

> ### Author Response · Authors · 2019-10-03
> **indeed**
>
> There was indeed an error in one of the trees, thank you for spotting it! We will fix it in the revised version of the paper.

---

### Public Comment · ~Nestor_Demeure1 · 2019-09-27
**Add independent test suite**

It would be interesting to include a test suite that was not generated using the same techniques as the training set to confirm that the method is able to generalize out of its training set.

A possibility would be the Rubi integration suite by Albert Rich: https://rulebasedintegration.org/

---

> ### Author Response · Authors · 2019-10-03
> **yes**
>
> Thank you for the reference! We are considering testing on alternative datasets, and the ones provided by Rubi seem indeed interesting as they come from a different distribution.

---

### Public Comment · ~Ronen_Tamari1 · 2019-09-27
**Training set details?**

Interesting paper!

Maybe I missed this somehow- what are the sizes of the training sets?

Also, it would be interesting to see train/test performance curves.

---

> ### Author Response · Authors · 2019-10-03
> **dataset size**
>
> Thank you for your comment!
>
> In practice, we use 40M equations for each task. We will add details about our datasets (number of equations, average number of nodes, operators, etc.) in the updated version of the paper.
>
> What we observe during training is that with smaller training sets the training accuracy is better (i.e. it is easier for the model to overfit on small training sets), but the model does not generalize as well and test accuracy is worse (which is typically what we observe in machine translation).

---

### Public Comment · ~S._Alireza_Golestaneh2 · 2019-09-28
**Can the code be available?**

Such an interesting work!
Can the code/implementation be available?

---

> ### Author Response · Authors · 2019-10-03
> **yes**
>
> Thank you for your message! Yes, we are planning to release the code after the review process.

---

### Public Comment · ~anima_anandkumar1 · 2019-09-29
**Previous work does a better job and this is completely missed**

This paper misses important papers that overlap significantly with the work.

The authors need to provide credit where due and closely compare this with the following works:
https://openreview.net/pdf?id=Hksj2WWAW
We did the first work that combined symbolic representation with function evaluation. We demonstrated the ability to extrapolate to higher depth than present in training data. In subsequent work, we also extended to differential equations:
https://uclmr.github.io/nampi/extended_abstracts/arabshahi.pdf

I don't see any new methodology beyond papers I mentioned above. In fact, they do worse: they don't do any extrapolation (testing beyond depth they trained on), and they only limited to symbolic evaluation.

---

> ### Author Response · Authors · 2019-10-03
> **Previous work does not solve differential equations, only checks them, and does not present techniques to generate ODE datasets**
>
> Thank you for these references. We will add them in the updated version of the paper.
>
> However, we respectfully disagree with your statements: the methodology and the tasks we tackle in our paper are very different from what you propose.
>
> First, we are working on very different tasks. You present an approach to check that a given function is a valid solution of a differential equation. This is a binary classification task, which is arguably a much easier problem than actually generating the solution from scratch, like we do.
>
> Second, your approach amounts to generating data by performing local random changes on a small set of mathematical expressions gathered from Wikipedia. An issue with this approach is that the resulting problem space is very localized and biased around the initial equations. In our case, we propose a sophisticated approach to generate random equations from scratch. Our generative process ensures that all trees are generated with the same probability over a very large space (Section 2, B and C of the appendix). This approach allows us to generate arbitrarily large expressions in a uniform way. We train and evaluate our model on expressions of up to 300 internal nodes, while your paper only considers equations with up to 15 internal nodes.
>
> The dataset in your paper is composed of 7000 differential equations, while our approach allows us to generate datasets that are several orders of magnitude larger. We train our models with 40000000 expressions, and could potentially use a lot more.
>
> Generalization to larger trees is important when the training set is restricted to small equations. We have no such restriction since we can generate equations of arbitrary depths. In our case, the generalization problem lies in the size of the problem space (Section B).
>
> The majority of studies in the field propose complex and dedicated architectures (like Tree-LSTMs) that are typically much slower and only applied to small datasets. One of the messages in our paper is that vanilla seq2seq models perform well on symbolic mathematics given enough data, and that dedicated architectures are not necessary. However, generating large datasets of differential equations is the real challenge, which was not addressed by previous works.

---

> > ### Public Comment · ~Forough_Arabshahi1 · 2019-10-03
> > **generalizability concerns**
> >
> > Thank you for your response.
> >
> > Using equations of a limited depth for training is indeed not a limitation of the data generation and rather is for the purpose of testing the model's generalizability performance to higher complexity beyond training data. Data of arbitrary size and depth can be generated using any reasonable automated data generation method. Of course if a model is over-saturated with data and only tested on data from the same distribution and domain, one will not be able to assess whether the model is just memorizing the data or is it actually learning to do something interesting.
> >
> > Moreover as shown in our paper and also other papers [1] (it seems like this reference was also not cited), [2] tree-structured models are the state of the art for symbolic math and logic and it is good practice to compare the performance of your proposed model with the state-of-the-art and back-up the claim that there is no need for a tree-structured model through experiments. In fact all the mentioned papers compare tree structured models against seq2seq models and show that tree-structured models outperform them (it might be worth mentioning that all models including seq2seq will be near perfect on data that is similar to training data). It is mentioned in your paper that you are focusing specifically on models used for NLP, however, in natural language the tree-structure is not inherent to the data itself and should be extracted using an external parser but in mathematics and logic the tree-structure is inherent to the data and ignoring it results in a loss in generalization performance. Vanilla seq2seq models might perfectly memorize the data but they will suffer in performance once the data becomes more complex (and if this is not the case, it will be nice to actually show it). Therefore, testing on larger trees or more complex datasets is not in any case only for problems that have access to small datasets, rather it is a test of model's generalizability.
> >
> >
> > [1] Richard Evans, David Saxton, David Amos, Pushmeet Kohli, Edward Grefenstette  "Can Neural Networks Understand Logical Entailment?", ICLR 2018
> > [2] Miltiadis Allamanis, Pankajan Chanthirasegaran, Pushmeet Kohli, and Charles Sutton. "Learning continuous semantic representations of symbolic expressions.", ICML 2017

---

> > > ### Author Response · Authors · 2019-10-06
> > > **regarding generalization**
> > >
> > > Thank you for your comment.
> > >
> > > You write: "of course if a model is over-saturated with data and only tested on data from the same distribution and domain, one will not be able to assess whether the model is just memorizing the data or is it actually learning to do something interesting". Although this is true for small problem spaces, it will not happen here. As we show in Section B of the appendix, there are over 1e11 expressions with five internal nodes, 1e23 with ten internal nodes, and 1e34 with 15 internal node. We use a training set of 4e7 equations with up to 15 internal nodes. Over-saturation with data in that setting is simply out of the question.
> > >
> > > To address your concern, we ran an additional experiment with a small model composed of 2 layers of dimension 128. It is clearly impossible for such a model to memorize a training set of 40M equations. With this model, we obtain an accuracy of 91.0% on a holdout test set, and 95.6% using a beam search of size 10. On a small subset of the training set, our model obtains the same accuracy, which shows that there was no overfitting, and that the model was properly able to generalize beyond the training set.
> > >
> > > We agree that it is important to compare with the SOTA. However, the SOTA in symbolic computation is held by computer algebra systems like Matlab and Mathematica, and not by neural tree-structured models. This is why we compare against these computer algebra systems. In fact, neural models (tree-structured or not) have never been tested on the tasks of function integration or differential equation solving (and not checking) before, so they cannot be the SOTA here. Tree architectures are discussed in the related work section of our paper. But as we explain, in mathematics they have mostly been used for arithmetic calculations (including logic) and for classification. This is a very different problem than what we are trying to solve.
> > >
> > > You write "Data of arbitrary size and depth can be generated using any reasonable automated data generation method.". Generating data of arbitrary size is not a trivial problem. The method you propose amounts to performing local random changes on a small set of mathematical expressions gathered from Wikipedia. We believe this is not a satisfactory method. As mentioned in our previous response, such a method will inevitably generate biased equations with expressions centered around initial equations. In our case, we propose an elaborated technique to generate unbiased expressions (Section C of the appendix) where all trees have the same probability of being generated.
> > >
> > > Besides, the general form for a n-th order differential equation in your paper is wrong: what you gave is the form for linear differential equations, not the general form. As a result, your approach can only generate linear differential equations which is again a simpler problem than the general case. Our approach (Sections 3.2, 3.3, C and D) presents a way to generate arbitrary differential equations of the first and second order, and to the best of our knowledge, such an approach has never been proposed before, in the machine learning or any other community.

---

> > > > ### Public Comment · ~Forough_Arabshahi1 · 2019-10-29
> > > > **Generalizability**
> > > >
> > > > Thanks a lot for your response!
> > > >
> > > > The main point of my previous comment, which is unanswered, was that the *test data* does not measure the generalizability of the model, since the test set has the same complexity as the training set. An example of a test set that can potentially measure generalizability performance is one containing deeper expressions than seen in training. There could also be other ways of measuring generalizability depending on the problem studied. You can e.g. take a look at the tests done in the Neural Turing Machine (NTM) [1], where they test the model's generalizability on the copy task by seeing the performance of NTM on sequences that are longer than the sequences in the train set. You can see there, too, that the model is doing very well on held-out data of the same complexity as train data. The interesting thing is to see how well the model generalizes to more complex datasets.
> > > >
> > > > In fact, your observation that a simple 2 layer network can achieve such a high accuracy on the held-out set could be a red flag that might mean the test set is too simple for measuring the model's true generalizability performance.
> > > >
> > > > If you are proposing a symbolic solver, then yes SOTA is a computer algebra system. But when you are proposing a neuro-symbolic solver, SOTA is the other neuro-symbolic solvers; although it is always nice to see comparisons with computer algebra systems. Specifically, there was a claim in your paper that tree-structured models are not needed. It is always good to back-up claims using experiments (and I note again that if the test data is of the same complexity as the train data, like it is here, this claim is probably true to some extent.)
> > > >
> > > > [1] Graves, Alex, Greg Wayne, and Ivo Danihelka. "Neural turing machines." arXiv preprint arXiv:1410.5401 (2014).
> > > >
> > > > P.S. Thanks for pointing us to the typo! We did not claim that we proposed a method for generating *differential equations*.

---

### Public Comment · ~Nick_Moran1 · 2019-10-03
**Two Points of Clarification**

Very interesting work, I have a few points which I feel are a bit unclear in the current version.

1) What is the space of output tokens that the model can emit?  Section 4.1 describes a set of tokens used to create the dataset, but this is restricted to numeric values in the range {-5,...,5}.  Presumably a similar restriction does not apply to the outputs of the model, given the examples with a '9' in Table 4.  In table 3, we see a solution with '14' as one the scalar values.  Is this emitted as a single token, as a concatenation of '1' and '4', or as an expression like '7 * 2' which is then simplified?

2) In sections 4.4 and 4.5, is accuracy calculated by checking whether the single most probable output found by beam search is correct, or if any of the top n outputs are correct?  The former seems like the natural way to evaluate the model, but this passage from section 6 seems to suggest that it may be the latter: "However, proposed hypotheses are sometimes incorrect, and considering multiple beam hypotheses is often necessary to obtain a valid solution. The validity of a solution itself is not provided by the model, but by an external symbolic framework (Meurer et al., 2017)."  If the latter, how often is the single most probable output correct?

---

> ### Author Response · Authors · 2019-10-06
> **re: Two Points of Clarification**
>
> Thank you for your questions!
>
> 1) The model has the same input and output space. A number like 14 is represented as "[INT+ ; 1 ; 4]" (i.e. by 3 tokens). The differential equation "y'-100=0" is represented as "[SUB ; Y' ; INT+ ; 1 ; 0 ; 0]" and the output will be "[ADD ; MUL ; INT+ ; 1 ; 0 ; 0 ; x ; c]" for "100x + c". So the model can receive and generate arbitrary integers. What we meant by the {-5 .. 5} generation range, is that in the initial representation of trees (before simplification), their leaves only have integer values in {-5 .. 5}. However, it is possible to have an initial expression like "y'-5*5*4=0" that will be simplified to "y'-100=0". This is why you can see examples in the paper with integers larger than 5. Adding this restriction in the generation allows us to reduce the size of the problem space, and to avoid having too many expressions with huge integers in the training set (these expressions can always be generated, but are less likely, and we found it useful as equations with huge integers are usually not very interesting or difficult to solve).
>
> 2) We actually do the later approach. In machine translation, the former approach (i.e. taking the most probable of the beam) makes indeed more sense as there is no clear way to verify the correctness of the translation. But in our case, since we can quickly verify a solution by plugging it into the equation it has to verify, we do the second approach and consider all hypotheses in the beam until we find a valid one. Evaluating how often the single most probable output is correct is interesting, and we did not try it before. We just tried for first order differential equations, and found that using a beam size of 10 and testing only the best hypothesis slightly improves the performance, but not by much (about 0.5% over beam size 1), which suggests that it is important at test time to explore more than one option.

---

> > ### Public Comment · ~Nick_Moran1 · 2019-10-07
> > **re: Two Points of Clarification**
> >
> > Thank you for your reply.  That all makes perfect sense.
> >
> > If the value of a wider beam search lies more in providing more plausible hypotheses than in merely maximizing the likelihood of the top output, I wonder if it might be possible to further improve accuracy using a different beam search heuristic.  For example, by trying to encourage a diversity of candidates rather than merely the top-k most probable.

---

> > > ### Author Response · Authors · 2019-11-09
> > > **beam search**
> > >
> > > You are absolutely right. Thank you for your suggestion. Encouraging diversity would probably allow to model to explore a wider set of candidates, and increase the probability to find a good solution. Actually, maybe a simple but effective solution could be to sample solutions instead of using a beam search. This is something we will investigate in the future.

---

### Public Comment · ~Bartosz_Piotrowski1 · 2019-11-07
**Generalization, similar work, other issues**

Thank you for an interesting work! I have several comments, though.

(1) Generalization. I very much agree with Forough Arabshahi that assessing generalization of the model is a crucial issue here. In my opinion it requires much more careful analysis. It could happen that the examples in the test set are in some way too similar to some (or many) instances in the training set, and the model does more memorization than the claimed generalization. I'm not saying this similarity is trivial, maybe it is some nuanced leak in the data, but it requires our attention, and this analysis can be interesting on its own. Which examples were problematic for the model? What characterizes the easy ones? Also, why did you use so small test set compared to 40M training set? Why accuracy is measured on 5000 examples but comparison with Mathematica is done only for 500 examples? At first glance it's a bit dubious. Releasing the data sets would be very appreciated in the context of these concerns.

(2) Similar work. Recently we did experiments which were in the same spirit -- but for easier and smaller data.  It was applying out-of-the-box seq2seq models for (a) normalizing polynomials (of varied complexity, generated synthetically) and (b) for learning rewriting steps extracted from automated proofs. (The work was presented at AITP'19 [1] and GNN workshop at ICML'19 [2]). In our experiments we also noticed, that prefix/Polish notation is helpful for applying NMT.

There is also another, earlier, work about NMT in symbolic setting where a problem being solved is translating informal LaTeX to formal math [3].

(3) TreeNNs. I understand the motivation for using classical seq2seq -- faster to train, performance is great. I believe, though, that research-wise it's important to not lose the focus on tree neural nets -- tree structure is intrinsic to symbolic expressions and its extraction is for free. TreeNNs can "directly comprehend" this tree structure. I believe this is the way to provide the right domain-specific architectural bias (like 2d convolutions is the right bias for images) and to achieve much more robust/controlled/explainable generalization. I hope to see advances in tree-based architectures for symbolic problems (even if these models are, initially, inferior efficiency/performance-wise.)

(4) Technical details. I would like to see more details such as: for how many epochs did you train the network, what hardware did you use for the evaluation with time-limited Mathematica, how the hyperparameters of the model were found (Transformer tends to be fragile with respect to hyperparameters.) This would be beneficial for increasing reproducibility.

[1] Piotrowski, Brown, Urban, Kaliszyk: Can Neural Networks Learn Symbolic Rewriting?, AITP 2019, http://aitp-conference.org/2019/aitp19-proceedings.pdf
[2] (title as above), GNN workshop at ICML 2019, https://graphreason.github.io/papers/40.pdf
[3] Qingxiang Wang, Cezary Kaliszyk, Josef Urban:
First Experiments with Neural Translation of Informal to Formal Mathematics. CICM 2018

---

> ### Author Response · Authors · 2019-11-09
> **Generalization**
>
> Thank you for your comment. We now address the generalization problem, please refer to the updated version of the paper (Sections 3.1, 4.4, E and F).
>
> A test set of 5000 is large enough to have a reliable estimate of the overall accuracy of our model. See Tables 4 and 5, where we evaluate our model on 500 and 5000 equations and obtain almost identical results.
> As already mentioned in the paper, the reason we considered 500 equations is because of the limited speed of the symbolic frameworks we considered. Besides, the difference of performance we observe between the models is already statistically significant for 500 equations.
>
> We agree that tree-structured models are an interesting alternative to seq2seq models. But although they are a natural choice for classification tasks, using them to transform an expression into another is more challenging. People have tried in the past to use tree-structured models for sequence generation in NLP, but with limited success compared to seq2seq models which remain the natural choice. We leave the study of applying tree-structured models to function integration and differential equation solving to future work.
>
> We actually found that our models were very stable, and that changing the architecture / learning rate scheduling had almost no impact on our results (please refer to our response to reviewer 1 for more details).
>
> As mentioned in the comments below, we will release our code and datasets for reproducibility.

---

### Author Response · Authors · 2019-11-09
**Updated version of the paper**

We thank the reviewers, and all participants in this discussion for their comments. They were extremely useful and helped us to improve the paper. To address them, we have been actively working on the paper since the initial submission. We just uploaded a new version with a lot of new results.

We replied to all reviewers and commenters individually. Below is a summary of the major changes between the updated version and the original submission:

1) Data generation for function integration has been deeply modified. To address the concerns about generalization, we now consider not 1 but 3 different generators to create functions with their integrals. Section 3.1 has been entirely rewritten, and describes our 3 generators in detail. In Section 4, we show that our model achieves excellent in-distribution performance on all three samples, and discuss out of distribution generalization. We believe this is the correct way to address generalizability issues, in self-supervised settings where datasets are generated.

2) A new section (Section E of the appendix) discusses generalization across datasets and studies differences between examples generated by our 3 generators.

3) A new section (Section F of the appendix) also addresses the generalization concern, and shows that a model trained to integrate exclusively functions that a symbolic framework (SymPy) can integrate, is able at test time to integrate functions that the symbolic framework is not able to integrate. This means that the model was able to generalize beyond the set of functions integrable by SymPy which it was trained on.

4) To address another concern about the timeouts, we added a new section (Section D of the appendix) experimenting with different timeouts for Mathematica. We show that the number of expressions on which Mathematica times out only represents a small fraction of the failure cases, and that Mathematica usually indicates that it cannot integrate the input equation before reaching the time limit. We also show that even in the ideal scenario where Mathematica would succeed on all equations where it times out, the difference in performance would remain small and this would not change the conclusions. We also conducted a test with Maple.

5) We added a graph at the end of Section 2 showing number of expressions and trees for different numbers of operators and nodes.

6) In Section 4, we added statistics about the data sets, such as the training set size, the average and maximum length of expressions, and the ratio between input and output lengths.

7) At the end of the evaluation section, we clarified how we use beam search, and that unlike in machine translation, we do not return the single hypothesis with the highest score, but that we consider all hypotheses in the beam.

8) In the appendix, we improved the algorithm for generating random expressions. The new algorithm produces the same distribution, but its derivation is clearer and it implementation cleaner.

9) We removed the alternate generator for second order ODEs, which was ultimately not needed.

10) At the end of the appendix, we added a page with examples of integrals generated by our three methods.


Finally, as many people have requested the code and datasets, we would like to confirm that we will release them after the review process.

---

### Decision · Program_Chairs · 2019-12-19

**Decision:**

Accept (Spotlight)

**Comment:**

The paper presents a deep learning approach for tasks such as symbolic integration and solving differential equations.

The reviewers were positive and the paper has had extensive discussion, which we hope has been positive for the authors.

We look forward to seeing the engagement with this work at the conference.